# Modeling of Multiphase Flow, Superheat Dissipation, Particle Transport, and Capture in a Vertical and Bending Continuous Caster

**Mingyi Liang [1]**, **Seong-Mook Cho [2]**, **Xiaoming Ruan [3]** and **Brian G. Thomas [1,*]**

[1] Department of Mechanical Engineering, Colorado School of Mines, Brown Hall W350, 1610 Illinois Street, Golden, CO 80401, USA; liang@mines.edu
[2] Department of Metallurgical Engineering, Pukyong National University, 45, Yongso-ro, Nam-gu, Busan 48513, Korea; smcho1@pknu.ac.kr
[3] Steelmaking Research Department, Research Institute Baoshan Iron & Steel Co., Ltd., 889 Fujin Rd., Shanghai 201900, China; ruanxm@baosteel.com
* Correspondence: bgthomas@mines.edu

**Abstract:** A new model of particle entrapment during continuous casting of steel is presented, which includes the effects of multiphase flow from argon gas injection and thermal buoyancy from superheat in the strand. The model simulates three different capture mechanisms, including capture by solidified hooks at the meniscus, entrapment between dendrites, and engulfment by the surrounding of large particles. The fluid flow and bubble capture results are validated with plant measurements, including nail board dipping tests and ultrasonic tests, respectively, and good agreement is seen. Results suggest that the superheat has a negligible effect on the flow in the mold region. However, higher (30 K) superheat causes a more complex flow in the lower strand by creating multiple recirculation zones due to the thermal buoyancy effects. This causes less penetration deep into the strand, which leads to fewer and shallower particle captures. Lower (10 K) superheat may enable significant top surface freezing, leading to very large internal defect clusters. Lower superheat also leads to deeper meniscus hooks, which sometimes (0.003%) capture large (1 mm) bubbles. Capture bands occur near the transition line from vertical to curved, due to the downward fluid velocity balancing the particle terminal velocity, enabling capture in the relative stagnation region beneath the longitudinal recirculation zone. These findings agree with plant observations.

**Keywords:** continuous casting; multiphase flow; superheat; meniscus hooks; bubbles; particle capture; steel slabs; computational flow model; CFD

## 1. Introduction

In the continuous casting of steel, fluid flow in the nozzle and strand can capture detrimental particles, such as inclusions from upstream ladle refining and tundish, mold slag entrainment, and argon bubbles, which are often injected into the nozzle in order to lessen nozzle clogging [1]. Although most of the particles escape through the top-surface slag layer, some can be captured beneath the meniscus hooks near the top surface [2,3]. Everywhere at the solidification front, small particles may flow in between primary dendrite arms and become entrapped by the solidifying steel. Large particles may become surrounded by growing dendrites to become "engulfed" in the solidifying shell. All of these captured particles can lead to permanent surface and internal defects, such as pinholes, slivers, blisters, and expensive rejects, or unsightly and dangerous quality problems in the final product [4]. Thus, there is a great incentive to understand the complex phenomena of particle transport and capture by multiphase fluid flow and solidification, to optimize the casting conditions, modify the nozzle geometries, and improve the steel quality.

Many researchers have investigated the transport and capture of argon bubbles and non-metallic inclusions by assuming instant capture when a bubble or inclusion touches

the solidification front referred to as the "simple criterion" [5–12]. Although this method is valid for particles smaller than the primary dendrite arm spacing (PDAS), it overpredicts the capture of particles too large to fit between the primary dendrite arms, which are often washed away back into the flow.

Efforts have been made to better predict the capture of bubbles larger than PDAS. Liu et al. [9] assume that bubbles with diameters larger than 500 μm are not captured by the solidification front, but this method naturally underpredicts the small but important capture fraction of such large bubbles. Liu and Li [13] assume that inclusions are captured by the solidification front where the liquid fraction is less than 0.6 [13,14]; moreover, Chen et al. [14] require the fluid speed to be less than 0.07 m/s. Thomas and coworkers [6,7,15–18] developed an advanced capture criterion, which includes both the entrapment of small particles and the engulfment mechanism of large particles, based on a force balance at the solidification front. This approach has been adopted [19,20] or modified [21] by others.

Results with the entrapment/engulfment criteria match well with plant step-milling measurements [16,19,20] and are better than with the simple criterion [16]. Two other studies have applied this entrapment/engulfment criteria for both argon bubbles and inclusions on both straight and bending sections of two plant casters [22,23]. A band of captured particles was predicted at around 1/4 and 3/4 across the slab thickness from the inner radius (IR) of the caster [14,21–23], which agrees with ultrasonic test (UT) maps of particle numbers and locations in as-cast slabs.

However, previous investigations with the entrapment/engulfment capture criteria underpredicted the bubbles captured near the slab surface, possibly due to the model neglecting the subsurface hook capture [16,24]. Hooks form during initial solidification at the meniscus and can degrade the steel quality, by entrapping argon bubbles and non-metallic inclusions beneath them while they move down, resulting in slab surface defects and slivers in the rolled product [2,25,26]. Several previous works investigated the effects of operating parameters on subsurface hooks [27–30]. Lee et al. developed an empirical equation to estimate the hook depth for ultra-low carbon steel [31], which confirms the expectation that increasing superheat delivery to the meniscus region or increasing the casting speed has a strong effect on decreasing the hook depth [31]. Recently, a hook capture mechanism was implemented into a computational fluid dynamics (CFD) model, by considering a bubble as captured if it stayed long enough in a hook zone [17]. Therefore, both this hook capture mechanism and the entrapment/engulfment capture criteria were implemented in the current study to predict argon bubble capture in the continuous casting process.

As another important phenomenon, superheat can potentially affect the particle capture by changing the molten-steel flow pattern in the strand region due to thermal and solutal buoyancy. Only a few previous studies have investigated superheat transport in computational flow models of continuous casting of steel [13,14,32,33]. These models show that most of the superheat is removed in the mold region, or just below [32,33]. The temperature of the molten steel in the mold decreases continuously with distance along the path traveled by the flowing jet, with the lowest temperatures found at the meniscus region, near the narrow faces (NFs), and near the submerged entry nozzle (SEN), for a typical double-roll flow pattern [32,33]. Huang et al. [33] show that the superheat has a negligible effect on the mold flow. Shi et al. [34] measured the temperature in a mold and showed that the Reynolds-averaged Navier–Stokes (RANS) model can match the measurements. However, no previous model has modeled fluid flow deep into the caster, where velocity is slower, and the importance of superheat is unknown. Superheat is known to affect the internal quality, owing to its strong effect on the microstructure. Lower superheat produces less centerline segregation and related quality problems [35–37]. However, its effect on particle capture has received little investigation.

To study the two-phase fluid flow of molten steel and argon gas in the nozzle and mold with different casting conditions and the subsequent impact on particle capture, many studies have been conducted using three-dimensional CFD models [9,12–14,16,19–21,23,24,33,38–45],

including Eulerian–Eulerian (E–E) [23,42–44] Eulerian–Lagrangian [9,12,14,16,19–21,23,24,38], and large Eddy simulation (LES) [13,14,24,39–41] approaches. It is important to include two-way coupling in multiphase flow simulations because the gas and molten steel flow affect each other's behavior. For instance, increasing the argon flow rate can change the mold flow from a classic double-roll to a single-roll flow pattern with the surface flow reversing to flow from the SEN towards the NF [43,44,46]. Excessive gas fractions have been observed to lead to oscillating and asymmetric flow [47,48]. Jin et al. developed a RANS k-ε model coupled with a discrete phase model (DPM) and the entrapment/engulfment capture criteria, to investigate the argon bubble capture in a continuous casting nozzle and a straight strand. The turbulent dispersion of each bubble can be successfully mimicked by the random walk model (RWM), which has been used in many previous studies [7–9,15,18,38,49–52]. Results show that 85 pct of small (<0.08 mm) bubbles are captured and only a small fraction of large bubbles is captured (<0.02 pct) [49].

Building on previous efforts [16,23], a new, computationally-efficient modeling methodology has been developed, to quantify the effect of superheat on the multiphase molten steel flow pattern and argon bubble capture in continuous slab casting. This new model system features an improved prediction of bubble size distributions and efficient post-processing to decrease the required number of injected bubbles.

## 2. Computational Models and Solution Procedure

The multiphase flow of molten steel with argon gas, superheat transport, and temperature distribution in the molten steel, particle transport, and particle capture via three different mechanisms was simulated using a system of computational models. Figure 1 shows a flow chart of the computational model system.

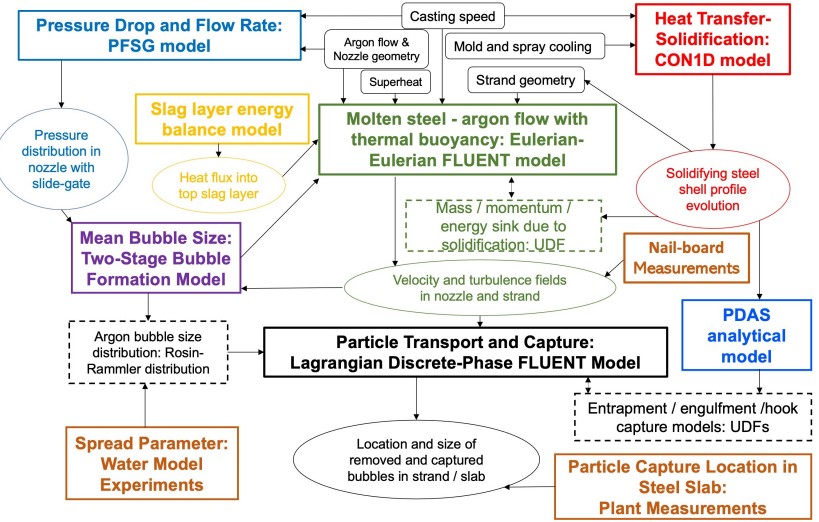

**Figure 1.** Model flow chart.

First, the mean bubble size needed by the multiphase flow model was predicted using a model of pressure distribution and flow rate within a slide-gate flow delivery system (PFSG) model [53] to find the local pressure, and a two-stage bubble size model for gas injection into downward flowing metal [54]. With the predicted mean bubble size and a spread parameter from the curve fitting of a water model experiment, the bubble size distribution was obtained. A heat transfer and solidification model (CON1D) was used to estimate the shell thickness profile down the strand, which was characterized with a solidification constant, and used to build the strand liquid pool geometry, implement the mass/momentum/energy sinks due to the steel solidification, and predict the PDAS size evolution.

Then, the multiphase flow in the nozzle and liquid pool inside the strand was computed with an E–E model (Ansys Fluent), including the effects of thermal buoyancy with

a fully-coupled heat transfer model of superheat transport. A simple energy balance model of the top slag layer was developed to calculate the heat flux lost from the top surface of the molten steel. Next, different bubble sizes were injected into the strand flow field and tracked using a Lagrangian DPM model (Ansys Fluent). This model includes hook/entrapment/engulfment capture mechanisms (fluent user-defined functions (UDFs)) to determine the capture of argon bubbles. A model of the PDAS size (down the caster) was needed to evaluate the capture mechanism. Finally, a new post-processing method was applied to calculate the particle capture fractions and compare them with measurements in the slab samples.

Details of all these computational models are provided in the following subsections.

*2.1. Initial Bubble Size Models: PFSG and Two-Stage Model*

The mean initial bubble size was calculated by the two-stage bubble formation model, knowing the pressure at the injection point from the PFSG model [53,54].

The PFSG model is a one-dimensional pressure energy model used to calculate the pressure distribution and flow rate in slide-gate nozzle systems for the argon–molten–steel flow system, including argon gas expansion by solving the Bernoulli equation [53]. This model was verified with CFD simulations and validated with plant measurements and water model measurements, with errors of less than 6% [53]. With the hot molten steel pressure $P_h$ at the injection location predicted by the PFSG model, the hot argon gas flow rate $Q_h$ can be calculated by:

$$Q_h = \frac{T_h P_s}{T_s P_h} Q_s \tag{1}$$

where $T_h$ is the casting temperature (K), $T_s$ is the standard condition temperature, 298.15 K, and $P_s$ is the standard condition pressure, 1 atm.

The two-stage bubble formation model of H. Bai was used to predict the mean argon bubble diameter $d_{p,mean}$, based on $Q_h$ [54]. During the first stage, argon bubbles expanded while holding onto the nozzle refractory pores. In the second stage, argon bubbles were detached from the nozzle wall and moved with the transverse liquid stream. The final bubble size $d_{p,mean}$ (at the time of detaching from the pore) was derived from certain detaching criteria based on experimental observations. The mean diameter, $d_{p,mean}$, was 5.3 and 4.9 mm for argon bubbles injected through the upper tundish nozzle (UTN) and the slide-gate upper plate, respectively.

Coalescence and breakup are important phenomena in the nozzle region where the gas is confined with a high volume fraction. Instead of modeling coalescence and breakup, as conducted by others [55], a Rosin–Rammler size distribution [56] was assumed for the argon bubbles exiting the nozzle. After entering the mold region, the bubble size distribution varies due to their different residence times [24]. However, a breakup is negligible, because the turbulence is so much lower, and coalescence is negligible due to the low gas fraction and rare bubble–bubble interactions. In addition, surface tension acts to keep most of the bubbles spherical.

The volume fraction of argon $F(d_{pi})$ contained in bubbles with diameters less than $d_{pi}$ is defined by the following Rosin–Rammler distribution [56]:

$$F(d_{pi}) = \frac{Q_h(d_p < d_{pi})}{Q_h} = 1 - e^{-\left(\frac{d_{pi}}{d_{p,mean}}\right)^n} \tag{2}$$

where the spread parameter, $n$, is taken as 3.5 based on the curve fitting of the bubble size measurements in a water model [57]. The bubble injection rate (#/s) through the UTN and slide-gate upper plate in the full-domain nozzle cross-section, $\dot{R}_i$, and the corresponding volume fraction of each bubble-size range are shown in Figure 2.

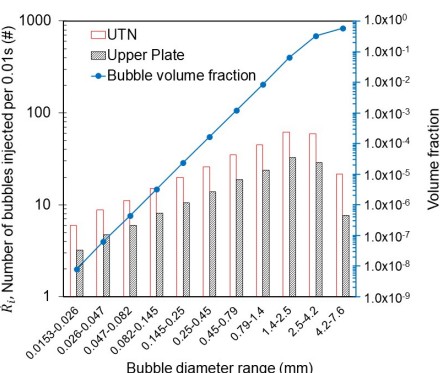

**Figure 2.** Injection rate and volume fraction of the bubbles.

### 2.2. Heat Transfer and Solidification of Solid Shell: CON1D

The shell thickness profiles in the strands on both the wide and narrow faces were calculated from the CON1D model [58]. This model includes a one-dimensional transient finite-difference calculation of heat transfer and solidification within the steel shell coupled with a steady-state model of heat conduction within the mold wall, which captures the effects of the water slots across the width direction using a reduced order model (ROM) [59]. CON1D extends the ROM to a two-dimensional mold model near the meniscus, to account for the vertical heat flux there. CON1D can predict the shell thickness, temperature distributions in the mold and shell, heat transfer across the interfacial gap, and other related phenomena [58]. Then the strand shell thickness can be roughly characterized by a solidification constant, $K$, of 3.2 mm/$\sqrt{s}$ (0.98 in/$\sqrt{min}$), based on fitting the CON1D output to:

$$s = K\sqrt{t_{cas}} = K\sqrt{\frac{l_c}{v_c}} \tag{3}$$

where $t_{cas}$ is the casting time (s), $l_c$ is the distance below the meniscus along the strand center (m), and $v_c$ is the casting speed (m/s).

Figure 3 shows that the empirical shell thickness compares reasonably with the plant data. In addition to defining the shape of the liquid pool for the model domain, the shell thickness profile from the empirical equation is implemented into UDFs for mass, momentum, energy sinks, and particle capture models.

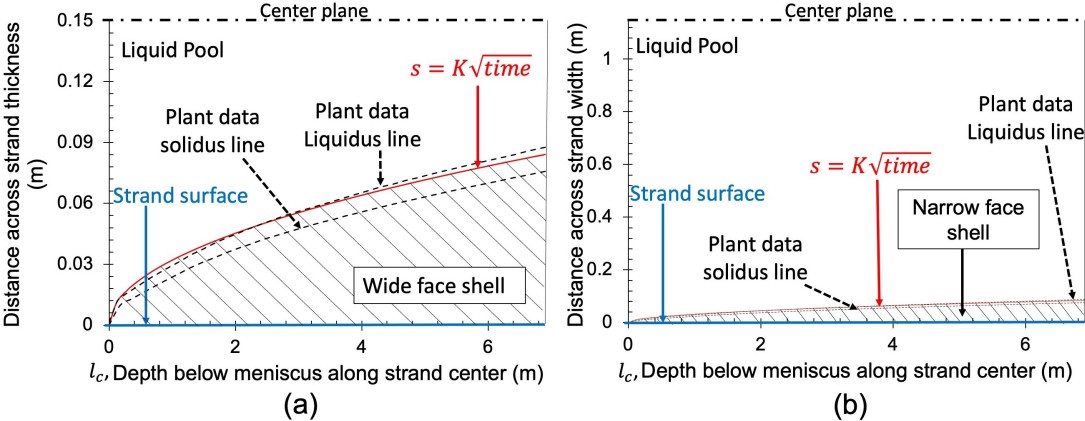

**Figure 3.** Shell thickness profile: (**a**) on the wide face (WF); (**b**) on the narrow face (NF).

### 2.3. Two-Phase Flow: Eulerian–Eulerian Model

A three-dimensional steady-state E–E RANS model was employed to simulate the multiphase flow of molten steel and argon gas in both the nozzle and the liquid pool of the

caster, using the commercial finite-volume software ANSYS FLUENT [60]. UDFs are used to account for mass, momentum, and energy sinks around the liquid pool boundary due to the steel solidification [6,61]. The governing equations of the E–E model solve separate continuity and Navier–Stokes equations for each phase. The continuity equation of each phase is as follows:

$$\frac{\partial}{\partial x_i} \alpha_q \rho_q u_{iq} = S_{q-mass-sink} \tag{4}$$

where the subscript $q$ is either $s$ or $g$, denoting the steel or argon gas phase, and the sum of the volume fraction of each phase is 1 in each cell ($\alpha_s + \alpha_g = 1$); $u_{iq}$ is the steel or argon gas velocity tensors; $\rho_q$ is the density of steel or argon; $S_{q-mass-sink}$ is the mass sink term for the liquid or gas phase, which is applied to the cells adjacent to the solidification front to account for the mass loss due to the liquid phase transitioning into the solid steel. The mass sink term for steel can be calculated by the equation below:

$$S_{s-mass-sink} = \rho_s v_c A_c V_{cell}^{-1} \tag{5}$$

where $A_c$ is the projected area of the wall-adjacent cell face in the casting direction (m$^2$), and $V_{cell}$ is the wall-adjacent cell volume (m$^3$). $A_c$ can be calculated by $A_c = A_f \times sin\theta_c$, where $A_f$ is the wall-adjacent face area (m$^2$), and $\theta_c$ is the angle between the face and the casting direction. Previous work calculates $\theta_c$ along the straight part of the strand [5–7,16,17,24,61]. Based on Equation (3), the following equation is derived for $\theta_c$ on the curved strand:

$$\theta_c = arcsin(\frac{\partial s}{\partial l_c}) = arcsin(0.5K\sqrt{\frac{v_c}{l_c}}) \tag{6}$$

The argon gas mass sink term is only applied to the top surface to simulate the process of argon gas escaping into the top slag layer, because the bubble diameter in the E–E model is set as the calculated mean bubble diameter $d_{p,mean}$, which is large. So, the continuous-phase bubbles tend to float up into the slag layer immediately after they flow through the nozzle port. The mass sink term for the argon gas can be calculated by the equation below:

$$S_{g-mass-sink} = \rho_g v_{gy} A_y \alpha_g V_{cell}^{-1} \tag{7}$$

where $v_{gy}$ is the gas phase vertical velocity towards the top surface, $A_y$ is the top-wall-adjacent cell face area (m$^2$), and $\alpha_g$ is the gas volume fraction in the top-wall-adjacent cell. The Navier–Stokes equations for momentum balance of each phase are:

$$\alpha_s \frac{\partial \rho_s u_{is} u_j}{\partial x_j} = -\alpha_s \frac{\partial P}{\partial x_i} + \frac{\partial}{\partial x_j}[(\mu_s + \mu_{ts})(\frac{\partial u_{is}}{\partial x_j} + \frac{\partial u_{js}}{\partial x_i})] + \alpha_s F_{Bi,g} - R_{igs} + \sum F_{is} + F_{Bi,t} + S_{is-momentum-sink} \tag{8}$$

$$\alpha_g \frac{\partial \rho_g u_{ig} u_j}{\partial x_j} = -\alpha_g \frac{\partial P}{\partial x_i} + \frac{\partial}{\partial x_j}[(\mu_g + \mu_{tg})(\frac{\partial u_{ig}}{\partial x_j} + \frac{\partial u_{jg}}{\partial x_i})] + \alpha_g F_{Bi,g} - R_{igs} + \sum F_{ig} + S_{ig-momentum-sink} \tag{9}$$

where $P$ is the pressure shared by all phases, and $\mu_s$ and $\mu_g$ are the dynamic viscosities of the molten steel and argon gas. The turbulent viscosities $\mu_{ts}$ and $\mu_{tg}$ are calculated by the standard $k$-$\epsilon$ model. The gas buoyancy force, $F_{Bi,g}$, caused by the density difference between the liquid and gas phase, is $(\rho_s - \rho_g)g_i$, where $g_i$ is the gravitational acceleration ($-9.81$ m/s$^2$ in $y$ direction). The thermal buoyancy force, $F_{Bi,t}$, is $(\rho_s(T) - \rho_0)g_i$, where $\rho_s(T)$ is the temperature-dependent density of the steel at a given location relative to the reference density, is discussed in the next section. The momentum sink terms for molten steel and argon gas, $S_{is-momentum-sink}$, and $S_{ig-momentum-sink}$, can be calculated based on their mass sink terms: $S_{iq-momentum-sink} = S_{q-mass-sink}u_{iq}$.

The additional forces on the liquid and gas phases, $\sum F_{is}$ and $\sum F_{ig}$, are defined as follows:

$$F_{is} = -F_{ig} = F_{iVM} + F_{iD} = \overbrace{0.5\alpha_g \rho_s (\frac{Du_{is}}{Dt} - \frac{Du_{ig}}{Dt})}^{\text{virtual mass force, } F_{iVM}} - \underbrace{\frac{3}{4}(\frac{C_{D1}}{d_p})\rho_s \alpha_g \mid u_{ig} - u_{is} \mid (u_{ig} - u_{is})}_{\text{drag force, } F_{iD}} \tag{10}$$

where $F_{iVM}$ is a virtual mass force, $F_{iD}$ is a drag force, $C_{D1}$ is a drag coefficient, and $d_p$ is the argon bubble diameter.

A source term for the interphase force, $R_{igs}$, is defined as:

$$R_{igs} = \alpha_g(1 - \alpha_g)\rho_g [\overbrace{\frac{C_{D1}Re_p}{24}}^{f_{drag}}](u_{ig} - u_{is})[\overbrace{\frac{\rho_g d_p^2}{18\mu_s}}^{\tau_p}]^{-1} \tag{11}$$

where $f_{drag}$ is a drag function, $\tau_p$ is the particle relaxation time that represents the time scale for a particle to respond to the changes in the surrounding flow, and $Re_p$ is the particle Reynolds number due to the relative difference between the particle and local fluid velocities, defined by $\frac{\rho_s|u_{ig}-u_{is}|d_p}{\mu_s}$. The Tomiyama drag model [62] was chosen. This model can be applied to a wide range of bubble-shaped regimes by including the Eotvos number $Eo$ in the drag coefficient ($C_{D1}$) calculation:

$$C_{D1} = max(min(\frac{24}{Re_p}(1 + 0.15Re_p^{0.687}), \frac{72}{Re_p}), \frac{8}{3}\frac{Eo}{Eo + 4}) \tag{12}$$

where $Eo = \frac{g(\rho_s - \rho_g)d_{p,eq}^2}{\sigma}$, which represents the ratio of buoyancy to the surface tension force, $\sigma$ is the surface tension (1.157 N/m) between argon gas and molten steel, and $d_{p,eq}$ is the sphere volume-equivalent bubble diameter.

For the turbulence modeling, the standard $k$-$\epsilon$ model was adopted [63]. The two-equation model attempts to predict the turbulence by solving equations for two variables, turbulent kinetic energy ($k$), and its dissipation rate ($\epsilon$). The model is given by

$$\frac{\partial\rho_m k}{\partial t} + \frac{\partial\rho_m u_{im}k}{\partial x_i} = \frac{\partial}{\partial x_j}[(\mu_m + \frac{\mu_{tm}}{\sigma_k})\frac{\partial k}{\partial x_j}] + G_{k,m} - \rho_m\epsilon \tag{13}$$

$$\frac{\partial\rho_m \epsilon}{\partial t} + \frac{\partial\rho_m u_{im}\epsilon}{\partial x_i} = \frac{\partial}{\partial x_j}[(\mu_m + \frac{\mu_{tm}}{\sigma_\epsilon})\frac{\partial\epsilon}{\partial x_j}] + \frac{\epsilon}{k}(C_{1\epsilon}G_{k,m} - C_{2\epsilon}\rho_m\epsilon) \tag{14}$$

where $G_{k,m}$ is the production of turbulent kinetic energy. The turbulent Prandtl numbers for $k$ and $\epsilon$, $\sigma_k$ and $\sigma_\epsilon$, are 1.0 and 1.3 respectively. Other constants have the following values: $C_{1\epsilon}$ = 1.44, $C_{2\epsilon}$ = 1.92, $C_\mu$ = 0.09.

The mixture density, $\rho_m$, mixture turbulent viscosity, $\mu_m$, and mixture velocity, $u_{im}$, are computed from $\rho_m = \alpha_g\rho_g + \alpha_s\rho_s$, $\mu_m = \alpha_g\mu_g + \alpha_s\mu_s$, and $u_{im} = \frac{\alpha_g\rho_g u_{ig} + \alpha_s\rho_s u_{is}}{\alpha_g\rho_g + \alpha_s\rho_s}$. The mixture turbulent viscosity $\mu_{tm}$ can then be calculated by $\mu_{tm} = \rho_m C_\mu \frac{k^2}{\epsilon}$. The turbulent viscosity $\mu_{ts}$ and $\mu_{tg}$ in Equations (8) and (9) can be calculated from $\mu_{tg} = \frac{\rho_g}{\rho_m}\mu_{tm}$ and $\mu_{ts} = \frac{\rho_s}{\rho_m}\mu_{tm}$.

### 2.4. Heat Transfer: CFD Thermal Model

The following steady, scalar, 3D energy equation is solved for the steel phase in the liquid pool that comprises the strand domain:

$$\frac{\partial\alpha_s\rho_s u_{is}h_s}{\partial x_i} = \frac{\partial}{\partial x_i}(k_s + k_{ts})\frac{\partial T_s}{\partial x_i} + S_{hs} \tag{15}$$

where $h_s$ is the sensible energy of the steel and can be calculated by $\int_{T_{ref}}^{T_{local,s}} c_{ps} dT_s$ $= c_{ps}(T_{local,s} - T_{ref})$, where $T_{ref}$ is the reference temperature, 298.15K, and $T_{local,s}$ is the predicted local temperature (K). The thermal capacity $c_{ps}$ and conductivity $k_s$ are in Table 1. The turbulent thermal conductivity, $k_{ts}$, can be calculated by $\frac{c_{ps}\mu_{ts}}{P_{rt}}$, where $P_{rt}$ is the turbulent Prandtl number, 0.85. The energy sink term, $S_{hs}$, is $S_{s-mass-sink}c_{ps}(T_{local,s} - T_{ref})$, and accounts for advection heat transfer across the solidification front.

The steel density is calculated [60] from the temperature results, based on its volumetric expansion coefficient $\beta$:

$$\rho_s(T) = \rho_0 \times [1 - \beta \times (T - T_0)] = 7000 \text{ kg/m}^3 \times [1 - 0.000100 \text{ K}^{-1} \times (T - 1826 \text{ K})] = 8278.2 - 0.7\,T \tag{16}$$

where $\rho_0$ is the density at the reference temperature, $T_0$. This density is needed to evaluate the thermal buoyancy term, $F_{Bi,t}$, in Equation (8). Solutal buoyancy would also affect the flow near the solidification front, where rejected solute from segregation may accumulate. In this work, solutal buoyancy was neglected, owing to the small segregation associated with the low carbon and low alloy content of the steels considered here.

*2.5. Energy Balance Model for Heat Flux into Top Slag Layer*

To obtain the heat flux boundary condition leaving the top surface of the strand model, an energy balance model of the top slag layer was developed. Solid slag is constantly added onto the molten steel to cover the top surface, serving as an insulating layer, aiming to prevent oxidation and freezing due to heat loss to the environment. The solid slag, after being added, gradually melts into a sintered state and liquid phase, forming a three-layer stratified flow. Under the steady-state, the energy coming in by adding the solid slag ($Q_{in,top}$) and the conduction from the bottom hotter steel ($Q_{in,bottom}$) has to equal that exiting by the convection from the top surface of the powder layer ($Q_{out,top}$) and the liquid slag consumed into the gap between the copper mold and steel shell ($Q_{out,bottom}$). These energy balances can be written as:

$$Q_{in,top} + Q_{in,bottom} = Q_{out,top} + Q_{out,bottom} \tag{17}$$

$$Q_{in,top} = c_{p,pow}(T_{top} - T_\infty)\rho_{slag}v_y A \tag{18}$$

$$Q_{out,top} = h_{sur}A(T_{top} - T_\infty) \tag{19}$$

$$Q_{out,bottom} = [\overbrace{c_{p,pow}(T_{slag,sol2} - T_{top})\rho_{slag}v_y}^{\text{powder layer heat flux}} + \overbrace{c_{p,sin}(T_{slag,liq} - T_{slag,sol2})\rho_{slag}v_y}^{\text{sintered layer heat flux}} + \overbrace{c_{p,liq}(T_{steel,liq} - T_{slag,liq})\rho_{slag}v_y}^{\text{liquid layer heat flux}}]A \tag{20}$$

where $A$ is the strand cross-sectional area and is the product of the strand thickness ($t_s$) and strand width ($w_s$). The heat flux from the hot molten steel into the liquid slag layer, $Q_{in,bottom}/A$, can be obtained by solving Equations (17)–(20). The heat capacities of the powder, sintered slag, and liquid slag, $c_{p,pow}$, $c_{p,sin}$, and $c_{p,liq}$, are 1000, 2750, and 3000 J/kg·K, respectively [64]. The ambient temperature, slag solidus temperature, and slag liquidus temperature, $T_\infty$, $T_{slag,sol2}$, and $T_{slag,liq}$ are 300, 1300, and 1400 K, respectively [64]. The powder top surface temperature, $T_{top}$, is 585 K, taken from trial 112 from A. Akhtar [64], and the top surface heat transfer coefficient to the environment, $h_{sur}$, is 2.7 W/m$^2$·K, which is averaged from trials 501 and 517 [64], because these trials have similar carbon content as the steel in this study. The slag density, $\rho_{slag}$, is taken as 2600 kg/m$^3$. The downward velocity of the slag, as it melts into liquid, as slag consumption, $v_y$, can be calculated by:

$$v_y = \frac{\overbrace{(c_{OM} + c_{lub}) \times f \times 2(t_s + w_s)}^{C_t}}{1000\rho_{slag}t_s w_s} \tag{21}$$

where $f$ is the estimated oscillation frequency, 0.93 cycles/s. The slag consumption rate is $C_t$ (g/s). The slag consumption rate per unit length per cycle due to the oscillation marks, $c_{OM}$ (g/m· cycle), and the slag consumption rate due to lubrication, $c_{lub}$ (g/m·cycle), can be calculated by the following empirical relations [65]:

$$c_{OM} = 2.5 \times 10^{-2} \times \rho_{slag} \times k_e^{1.43} \times t_n^{0.389} \times v_c^{-1.49} (\sqrt{\frac{2\sigma_{ss}}{\triangle \rho g}})^{0.556} \tag{22}$$

$$c_{lub} = 0.507 \times e^{3.59 \times t_p} \tag{23}$$

where $k_e$ is an empirical constant, 17.8; $\sigma_{ss}$ is the surface tension between slag and steel, 1.3 N/m; $\triangle \rho$ is the density difference between steel and slag, 4527 kg/m$^3$. The negative and positive strip times, $t_n$ and $t_p$, are calculated from: $t_n = \frac{1}{\pi f} cos^{-1}(\frac{v_c}{\pi s_0 f})$ and $t_p = T_C - t_n$, where $s_0$ is the oscillation stroke (7 mm) and $T_C$ is the total cycle time (s) [64].

To solve this model, Equations (21)–(23) are first solved for $v_y$. Then, Equations (17)–(20) are solved for the heat flux exiting the molten steel into the top slag layer: $Q_{in,bottom}/A$ is 164.7 kW/m$^2$ for both 10 and 30 K superheat cases.

### 2.6. Particle Transport: Lagrangian Discrete Phase CFD Model

Particles are injected into the flow field calculated with the E–E model and tracked by a DPM model. The following momentum balance equation is solved for each bubble with mass $m_p$ and velocity $u_{ip}$:

$$m_p \frac{Du_{ip}}{Dt} = \underbrace{\frac{m_p 18\mu_s}{\rho_g d_p^2} \frac{C_{D2}}{24} (u_{is} - u_{ip}) \overbrace{\frac{\rho_s d_p \mid u_{ip} - u_{is} \mid}{\mu_s}}^{Re_p}}_{F_{iD}} + \underbrace{0.5 m_p \frac{\rho_s}{\rho_g} (\frac{Du_i}{Dt} - \frac{Du_{ip}}{Dt})}_{F_{iVM}} + \underbrace{m_p \frac{\rho_s}{\rho_g} \frac{Du_{ip}}{Dt}}_{F_{iP}} + \underbrace{m_p \frac{g_i(\rho_g - \rho_s)}{\rho_g}}_{F_{iB}} \tag{24}$$

where $F_{iP}$ and $F_{iB}$ are the pressure gradient force and buoyancy/gravity force. The drag coefficient $C_{D2}$ was computed based on the drag law proposed by Moris et al. [66]:

$$C_{D2} = a_1 + \frac{a_2}{Re_p} + \frac{a_3}{Re_p^2} \tag{25}$$

$$a_1, a_2, a_3 = \begin{cases} 0, 24, 0. & Re_p < 0.1 \\ 3.690, 22.73, 0.0903 & 0.1 < Re_p < 1 \\ 1.222, 29.1667, -3.8889 & 1 < Re_p < 10 \\ 0.6167, 46.50, -116.67 & 10 < Re_p < 100 \\ 0.3644, 98.33, -2778 & 100 < Re_p < 1000 \\ 0.357, 148.62, -47500 & 1000 < Re_p < 5000 \\ 0.46, -490.546, 578700 & 5000 < Re_p < 10,000 \\ 0.5191, -1662.5, 5416700 & Re_p > 10,000 \end{cases} \tag{26}$$

During the DPM particle trajectory tracking procedure, the isotropic RWM was used to approximate the chaotic dispersion of particles due to turbulence. In this model, a Gaussian-distributed random velocity fluctuation, $u_i'$, is generated ($u_i' = \xi\sqrt{2k/3}$), which assumes isotropic turbulence everywhere in the fluid domain, in proportion to the local turbulence level. $\xi$ is a standard normally-distributed random number (i.e. mean = 0 and standard deviation = 1). The random number is not changed until the interaction time of the eddy and an Ar bubble $t_{inter}$ first reaches the eddy lifetime $t_e$ or the eddy cross-time $t_{cross}$ (the time required for an Ar bubble to pass the eddy). These two time scales are defined as: $t_e = -0.15\frac{k}{\epsilon} \ln \gamma$, and $t_{cross} = -\tau_p \ln [1 - \frac{L_e}{\tau_p|u_{is} - u_{ip}|}]$, where $\gamma$ is a uniformly-distributed random number from 0 to 1, and $L_e$ is the eddy length scale.

In this one-way coupled DPM model used here, particle–particle interactions are rare in the strand, and so are ignored. This makes particle injection and tracking very easy and flexible in this model system.

### 2.7. Particle Capture Model: FLUENT UDFs

Equations describing the hook, entrapment, and engulfment capture mechanisms are coded into UDFs and were used in this study to determine the capture of argon bubbles. The flow chart of this capture process is shown in Figure 4.

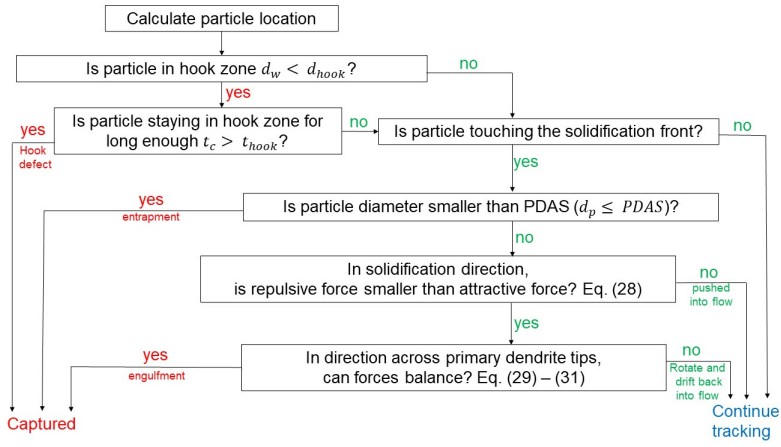

**Figure 4.** Capture-mechanism flow chart.

Particles can be captured as they rise beneath the solidified meniscus hooks, which are modeled by the hook capture mechanism [17]. The hook zone is defined as a region below the hook and above the solidifying shell, as shown in Figure 5a. There are three possible fates of a particle entering the hook zone: (1) touch the moving-down hook and become captured, (2) touch the solidification front and evaluate the criteria for the entrapment/engulfment mechanisms, and (3) flow out of the hook zone and back into the bulk flow.

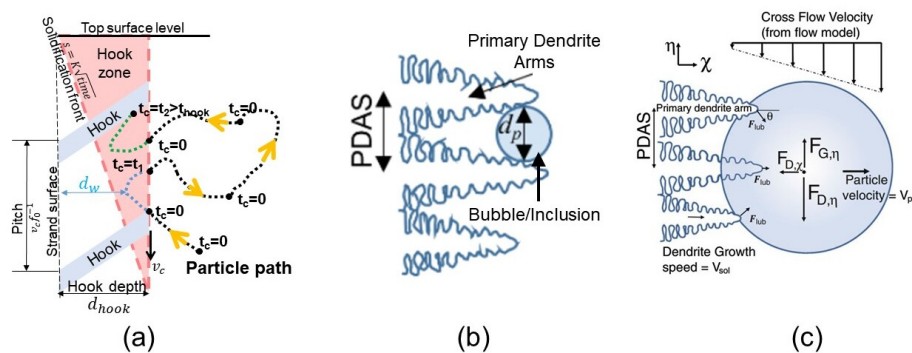

**Figure 5.** Capture mechanism schematic: (**a**) capture by a downward-moving hook; (**b**) entrapment in between primary dendrites arms; (**c**) engulfment by a growing solidification shell.

The hook depth $d_{hook}$ is defined as the horizontal distance between the hook tip and the slab surface. Lee et al. conducted experiments on ultra-low carbon steel samples to quantify the relation between the casting parameters and the hook depth, and an equation was proposed to predict the hook depth as a function of mold oscillation frequency $f_0$, mean level fluctuation $L_F$, and solidification temperature of slag powder $T_{slag,sol1}$ [31]. In this study, this equation was applied for low-carbon steel with $f_0$ = 68 cycles/min, $L_F$ = 1.8 mm, and $T_{slag,sol1}$ = 1150 °C. The results are hook depths of 5.6 mm with low (10 K) superheat and 4.5 mm with high (30 K) superheat. The real hook depth was rarely this deep, possibly because the empirical equation was developed from experiments on ultra-

low carbon steel, which is more susceptible to hook formation. However, the calculated deep hooks can be used to investigate worst-case scenarios regarding hook capture and compare with the entrapment/engulfment mechanisms to study the importance of the hook capture mechanism.

To prevent the solidifying shell from sticking to the mold, the mold oscillates at a frequency. Each oscillation cycle allows a new meniscus hook to form, and the shortest distance between every two adjacent hooks (pitch), $\Delta h_{hook}$, can be calculated by: $\Delta h_{hook} = v_c f_0^{-1}$. Then $\Delta h_{hook} = 0.0100 \text{ m/s} \times \frac{68 \text{ cycles}}{60 \text{ s}} = 11.3$ mm.

When the distance between a particle and the strand surface $d_w$ is less than $d_{hook}$, the particle enters the hook zone. The hook capture mechanism assumes that if a particle stays long enough in the hook zone, it will be captured. The critical time for a particle to be captured, $t_{hook}$, is approximated as the time required for a particle to travel half of a pitch:

$$t_{hook} = \frac{\Delta h_{hook}}{2v_c} = 0.5 f_0^{-1} \tag{27}$$

In this study, $t_{hook} = 0.44$ s. A cumulative timer $t_c$ is set up to track the time that each particle entering the hook zone stays there, as shown in Figure 5a. If $t_c > t_{hook}$, the particle will be considered as captured by the hook. When a particle is not in or escapes the hook zone, $t_c$ is reset to 0 s.

The entrapment/engulfment capture mechanisms [5,7,16–18,22–24] are applied to predict the capture of particles touching the solidification front. Particles smaller than the PDAS can flow between arms and become entrapped, as shown in Figure 5b. Larger particles may become engulfed in the solidification front, if the particle remains stationary, i.e., the forces cannot rotate the particle around the dendrite tips, as shown in Figure 5c. The latter occurs according to a balance of six different forces at the solidification front: drag force ($F_D$), buoyancy/gravity force ($F_B$), lift force ($F_L$), lubrication force ($F_{Lub}$), van der Waals force ($F_I$), and surface tension gradient force ($F_{Grad}$).

If the forces acting on a particle that touches the solidification front do not satisfy Equation (28), meaning that the net force pushes away in the solidification direction, the particle is pushed back into the flow and can be continuously tracked.

$$F_L - F_{B,\chi} - F_{D,\chi} - 2(F_{Lub} - F_{Grad} - F_I)cos\theta < 0 \tag{28}$$

where $\chi$ denotes the solidification direction, and $\theta$ is defined as: $\theta = arcsin(\frac{PDAS}{2(R_p + r_d)})$, with $R_p$ and $r_d$ as the particle radius and the dendrite tip radius. Otherwise, if Equation (28) is satisfied, the next step is to check if the forces in $\eta$ direction (tangential to solidification direction) can rotate the particle around the primary dendrite tips by the following equations:

if $F_{D,\eta}$ and $F_{B,\eta}$ are in the same direction:

$$(F_{D,\eta} + F_{B,\eta})cos\theta + (F_L + F_{D,\chi})sin\theta < (F_{Lub} - F_{Grad} - F_I)sin2\theta \tag{29}$$

if $F_{D,\eta}$ and $F_{B,\eta}$ are in the opposite direction, and $F_{D,\eta} \geq F_{B,\eta}$:

$$(F_{D,\eta} - F_{B,\eta})cos\theta + (F_L + F_{D,\chi})sin\theta < (F_{Lub} - F_{Grad} - F_I)sin2\theta \tag{30}$$

if $F_{D,\eta}$ and $F_{B,\eta}$ are in the opposite direction, and $F_{D,\eta} < F_{B,\eta}$:

$$(F_{B,\eta} - F_{D,\eta})cos\theta + (F_L + F_{D,\chi})sin\theta < (F_{Lub} - F_{Grad} - F_I)sin2\theta \tag{31}$$

If any of Equations (29)–(31) are satisfied, the particle rests on the solidification front and becomes engulfed. Otherwise, the particle rotates and drifts back into the flow.

The buoyancy/gravity force ($F_B$) has the same definition as in Equation (24). The drag force that points tangentially against the movement of a particle is defined as:

$$F_D = \frac{1}{8}\pi\rho_s d_p^2 C_{D3}|u_s - u_p|(u_p - u_s)$$ (32)

where $C_{D3}$ is the drag coefficient, defined as $C_{D3} = f_{Re_p}(\frac{24}{Re_p})$. $f_{Re_p}$ is the correction factor due to a finite particle Reynolds number [67]: $f_{Re_p} = (1 + 0.15Re_p^{0.687})$.

The lift force, caused by the shear flow, is calculated by [68]:

$$F_L = \frac{-9}{\pi}\mu_s R_p^2(u_s - u_p)sgn(G)\left[\frac{|G|}{v_s}\right]^{\frac{1}{2}} J^u$$ (33)

where $v_s$ is the molten steel kinematic viscosity, $G$ is the wall-normal velocity gradient, and $J^u$ is determined by a dimensionless parameter, $\varepsilon$, defined as $\varepsilon = sgn(G)\frac{\sqrt{|G|v_s}}{u_s - u_p}$. $J^u$ can be determined by the following equation when $0.1 \leq \varepsilon \leq 20$ [69]:

$$J^u = 0.6765\{1 + tanh[2.5log_{10}\varepsilon + 0.19]\}\{0.667 + tanh[6(\varepsilon - 0.32)]\}$$ (34)

For other $\varepsilon$ values, $J^u = -32\pi^2[sgn(\varepsilon)\varepsilon]^5 log\frac{1}{\varepsilon^2}$

The lubrication force is caused by the molten steel moving into the gap between the particle and the dendrite tip. This force can be written as [70,71]:

$$F_{lub} = 6\pi\mu_s V_{sol}\frac{R_p^2}{h_o}\left(\frac{r_d}{r_d + R_p}\right)^2$$ (35)

where $V_{sol}$ is the solidification velocity, and $h_o$ is the distance between the dendrite tip and the particle. The van der Waals force is calculated by [72]:

$$F_I = 2\pi\Delta\sigma_o\frac{r_d R_p}{r_d + R_p}\frac{a_o^2}{h_o^2}$$ (36)

where $\Delta\sigma_o = \sigma_{sp} - \sigma_{sl} - \sigma_{pl}$, with $\sigma_{sp}$, $\sigma_{sl}$, and $\sigma_{pl}$ as the surface tensions between the solid shell and particle, solid shell and liquid steel, and particle and liquid steel, respectively; $a_o$ is the atomic diameter of molten steel.

The surface energy gradient force is caused by the rejection of the solute near solidification front, mainly sulfur. The net surface tension force pushes the particle towards the solidification front and can be written as [6,73]:

$$F_{Grad} = -\frac{m\beta_1\pi R_p}{\xi_1^2}\left\{\frac{(\xi_1^2 - R_p^2)}{\beta_1^2}ln\left[\frac{(\xi_1 + R_p)[\alpha_1(\xi_1 + R_p) + \beta_1]}{(\xi_1 - R_p)[\alpha_1(\xi_1 + R_p) + \beta_1]}\right] + \frac{2R_p}{\alpha_1}\right\} + \frac{m\beta_1^2\pi R_p}{(\xi_1\alpha_1)^2}ln\left[\frac{\alpha_1(\xi_1 + R_p) + \beta_1}{\alpha_1(\xi_1 - R_p) + \beta_1}\right]$$ (37)

where $\alpha_1 = 1 + n_1C_o$, $\beta_1 = n_1r_d(C^* - C_o)$, $\xi_1 = R_p + r_d + h_o$. The empirical constants, $m$ and $n_1$, are 0.17 J/m$^2$ and 844 (mass%)$^{-1}$, respectively. $C^*$ can be calculated by $\frac{V_{sol}r_d}{D_s} = \frac{C^* - C_o}{C^*(1 - k_d)}$, where $C_o$ is the sulfur content of steel, $D_s$ is the diffusion coefficient of sulfur in steel, and $k_d$ is the distribution coefficient. Details of these criteria and forces are provided elsewhere [6,7].

Particles exiting the domain outlet and remaining in the domain after 500 s of the simulation time are considered captured as well, because the domain is 7-m long, and it is unlikely for a bubble traveling that deep and long to float up and be removed by the top slag layer.

## 2.8. Primary Dendrite Arm Spacing

The particle capture model requires the local PDAS profile, which increases with the distance below the meniscus due to the changing solidification rates and temperature

gradients at the solidification front. Several empirical correlations exist in the literature, based on measurements for different steel grades and cooling rates [58,74–77].

The dendrite arm spacing increases with lower carbon content, as shown by various researchers [75,76,78]. As presented previously [23], the PDAS for low-C steel can be predicted as follows, based on previous literature [6,79]:

$$PDAS = 4.3\lambda^{\frac{1}{4}} V_{sol}^{-\frac{1}{4}} G_T^{-\frac{1}{2}} \tag{38}$$

where the constant, $\lambda$ is $7.2 \times 10^{-13}$ m$^3$ °C$^2$/s and $4.3 \times 10^{-13}$ m$^3$ °C$^2$/s for the wide face (WF) and narrow face (NF), respectively, based on measurements [23]. The solidification front velocity, $V_{sol}$ (m/s), and the temperature gradient at the solidification front, $G_T$ (°C/m), can be extracted from the solidification constant $K$, the steel liquidus temperature $T_{steel,liq}$ (°C), steel thermal diffusivity $\alpha_{th}$ (m$^2$/s), and casting time $t_{cas}$ (s) according to classic analytical solutions for solidification [80]: $V_{sol} = \beta_{th}\sqrt{\frac{\alpha_{th}}{t_{cas}}}$ and $G_T = \frac{T_{steel,liq} + \triangle T_{sup} - T_{int}}{e^{\beta_{th}^2} erf(\beta_{th})\sqrt{\pi \alpha_{th} t_{cas}}}$, where $T_{int}$ is the interfacial temperature between solid and liquid steel (1000 °C), $\triangle T_{sup}$ is the molten steel superheat (K), $\alpha_{th} = \frac{k_s}{\rho_s c_{ps}}$, and $\beta_{th} = \frac{K}{2\sqrt{\alpha_{th}}}$.

This relation for low-C steel lies between other empirical equations [75,76] and previous measurements [76–78,81].

### 2.9. Flow Model Domain and Mesh

The flow model domain, including the complete slide-gate opening, nozzle, and 7-m-long liquid pool in the strand is shown in Figure 6. Due to its symmetry about its center plane, only half of the flow domain is modeled. The caster has a 2.665 m vertical upper section, followed by bending. To construct the liquid pool domain, multiple cross-sectional planes (perpendicular to the casting direction) were generated at different distances below the meniscus along the strand, with more planes created near the meniscus. The liquid pool width-by-thickness sketches were built on each plane based on the shell thickness profile in Figure 3, then the liquid pool domain was created by connecting all of the two-dimensional sketches using the ANSYS Workbench Design Modeler. In this way, the need to model the solidifying steel shell was avoided, and the domain shape was constant throughout the simulation.

A mesh of structured hexahedral cells was created in the ANSYS Workbench meshing and included ~20,000 cells for the nozzle domain and ~1.1 million cells for the liquid pool in the strand. Numerical diffusion increases when the mesh has an angle relative to the flow direction and is worst for 45° [82], so the mesh was constructed to align cell faces parallel to the nozzle jet flow direction to minimize numerical diffusion. Convergence for the steady-state simulation on a wide caster (width > 2 m) is difficult because the nozzle jet must travel further to reach the NF, which destabilizes the flow for this wide (2.3 m) caster, so first-order upwind was used for discretization of the advection terms in Equations (8), (9) and (13)–(15). The standard wall function was used for flow in the wall-boundary cells [63]. This allowed a coarser mesh near the boundaries, due to its reasonable treatment of the turbulent boundary layer. The maximum boundary cell size in the strand of the current mesh has a Y$^+$ value of less than the recommended maximum of 300 [60].

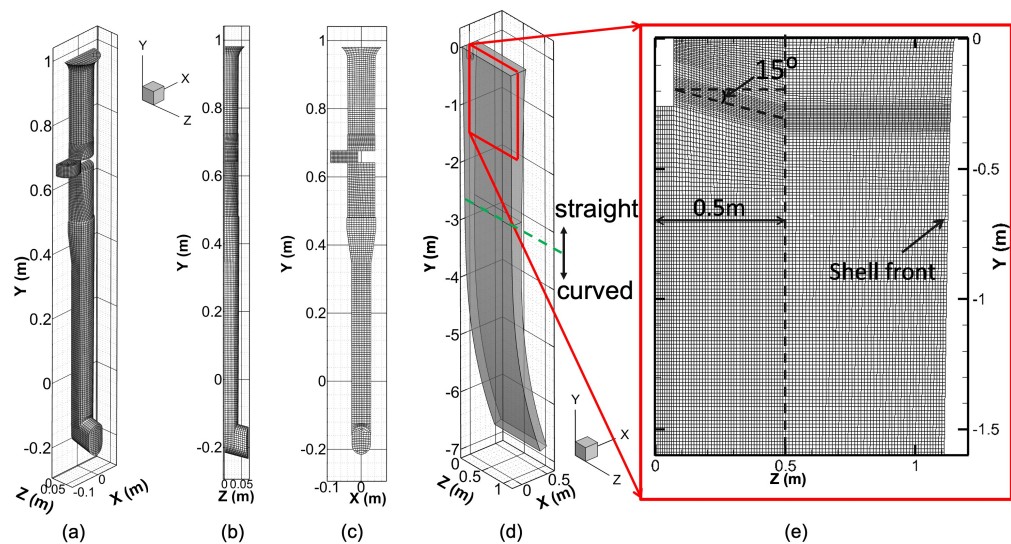

**Figure 6.** Flow domain and mesh: (**a**) nozzle 3D view, (**b**) front view, (**c**) nozzle center plane, side view, (**d**) liquid pool, and (**e**) liquid pool center plane.

Figure 6a–c shows the nozzle domain mesh. Figure 6d,e shows the mold domain geometry and the upper 1.6-m-high mesh on the center plane. The mesh near the mold inlet is refined due to the finer mesh on the nozzle outlet and the 1-on-1 data interpolation of velocity and turbulence from the nozzle outlet to the mold inlet. Within 0.5 m from the symmetry plane, the mesh was constructed with a 15° downward angle to help align the mesh with the flow direction, and thereby reduce false diffusion. Beyond 0.5 m towards the NF, the jet tends to lose its downward momentum and starts to flow horizontally. The mesh was then made parallel to the $z$ direction until the steel shell front.

### 2.10. Boundary Conditions and Solution Methodology

The boundary conditions for nozzle flow (step 1), isothermal strand flow (step 2), multiphase strand flow with superheat (step 3), and bubble transport and capture (step 4) simulations are given in Figure 7.

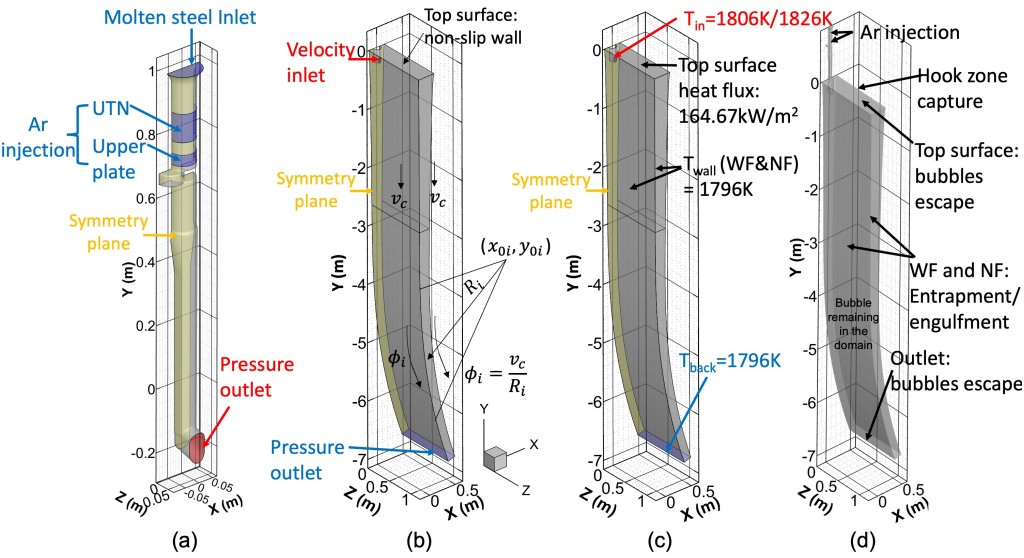

**Figure 7.** Boundary conditions for (**a**) step 1: nozzle flow, (**b**) step 2: strand flow, (**c**) step 3: heat transfer, and (**d**) step 4: bubble transport and capture.

Step 1 is to obtain the isothermal flow in the nozzle. Because of the short residence time of molten steel in the nozzle ($\frac{nozzle\ length}{avg.velocity} = \frac{1.2\ m}{1.4\ m/s} = 0.86$ s), the heat transfer in the nozzle is negligible. Then, the nozzle outlet velocity, turbulent kinetic energy, and turbulent dissipation rate are interpolated onto the inlet plane (nozzle port) (Figure 7b), and in step 2, the isothermal flow is calculated in the strand. In step 3, the temperature field is first estimated based on the steady-state isothermal flow pattern from step 2. Then the steel density is updated based on this initial temperature field and a fully-coupled temperature-multiphase fluid-flow calculation is performed to find the final E–E flow field. Finally, in step 4, discrete argon bubbles are injected into the E–E flow field from step 3, and bubble capture is determined with the capture models described in Section 2.7.

In step 1, a constant mass flow rate of 24.15 kg/s is applied at the inlet; the turbulent kinetic energy $k$ and its dissipation rate $\epsilon$ are fixed at $10^{-5}$ m$^2$/s$^2$ and $10^{-5}$ m$^2$/s$^3$. The argon gas volume flow rate and mean bubble size are used for the argon gas injection through both the UTN and upper plate, at the locations shown in Figure 7a. The argon gas is injected at 0.015 m/s through the UTN and 0.016 m/s through the upper plate, based on the hot argon volume flow rate divided by the inlet area. The nozzle wall has 0.00100 m roughness and has a non-slip boundary condition.

In steps 2 and 3, the top surface is approximated as non-slip for the molten steel flow, due to the large viscosity of the slag layer. Other surfaces (the WF and NF) are also set as non-slip walls with the mass and momentum sink UDFs acting in their adjacent cells. The thermal model in step 3 has an energy sink UDF. The narrow and wide faces are moving in the casting direction at the casting speed. As shown in Figure 7b, to simulate this wall velocity in the ANSYS FLUENT, the straight part of the narrow and wide faces are given a vertical velocity 0.0100 m/s (casting speed) in the casting direction. The curved part is given a rotational velocity, $\phi_i$ (rad/s), about the arc center ($x_{0i}$, $y_{0i}$), based on the curved arc radius $R_i$ (m) and the casting speed $v_c$ (m/s), where i = 1, 2, ..., 7, as there are 7 arcs with different radii: $\phi_i = \frac{v_c}{R_i}$. A pressure outlet condition is chosen over the nozzle outlet and the strand domain bottom, according to the ferrostatic pressure head of the molten steel ($\rho_s g h$), where $h$ is the vertical distance below the meniscus. The turbulent kinetic energy and its dissipation rate are set as $k = 10^{-5}$ m$^2$/s$^2$ and $\epsilon = 10^{-5}$ m$^2$/s$^3$ for the reversed flow (in case the flow re-enters the domain) throughout steps 1–3. In step 3 (Figure 7c), the inlet temperatures are 1806 K (10 K superheat) and 1826 K (30 K superheat). The top-surface heat flux into the liquid slag layer is 164.7 kW/m$^2$. The narrow face and wide faces, representing the solidification front, are set at the steel liquidus temperature, 1796 K, as is any flow re-entering the domain. Throughout steps 1–3, the gradient of every variable on the center symmetry plane is 0.

Finally, in step 4, the domains and flow patterns in the nozzle (step 1) and the strand (step 3) are combined, and the trajectories of discrete bubbles are then computed by the DPM model in a one-way-coupled manner. A certain number of bubbles are injected for each of the 11 bubble sizes in Figure 2. As its trajectory is integrated, each injected bubble can either escape into the top surface (slag layer) or be captured by the hook mechanism, entrapment, or engulfment in the hook zone, the vertical part of the strand, or later, deep in the curved strand. Particles that either remain in the domain after 500 s of simulation or exit the domain outlet are assumed to be captured deep in the strand, as shown in Figure 7d. The time step size is 0.005 s to satisfy the Courant number criterion.

### 2.11. Numerical Details

All the governing equations were discretized and solved by a Gauss–Seidel linear equation solver and an algebraic multigrid (AMG) method in ANSYS FLUENT with its coupled algorithm [60], on a staggered grid with zero velocities everywhere as initial conditions. This algorithm simultaneously solves the equations of velocity corrections of each phase and pressure correction [83] and is very efficient in steady-state simulations [60].

The simulation of the nozzle flow (Step 1) was conducted on a Dell computer tower with Intel (R) Xeon (R) E5-2609 v3 @ 1.90 GHz CPU, 64.0 GB RAM, and 10 computational

nodes. The nozzle simulation took ∼30 min, with all residuals reaching $10^{-4}$. For the strand flow (steps 2 and 3), a high-performance computer Mio was used with 48 computational nodes, and the results were considered converged when all residuals reached $10^{-7}$. The computational time for the mold flow was ∼5 h. The same computational power was used for step 4, and each transient simulation of 500 s took ∼1 day. After 500 s, more than 99.98% of the bubbles ended up either escaping into the top slag layer, exiting the outlet, or being captured by hooks, entrapment, or engulfment.

## 3. Post-Processing of Bubble Capture

When running the model to predict particle capture, transient simulations should be conducted for sufficient time to obtain statistically significant capture at every location and particle size. However, particles can be captured at a given location of the caster for a long time, because they can penetrate faster than the casting speed, and can persist in the strand for many residence times. The average residence time in the 7-m-long computational domain can be estimated by:

$$t_r = \frac{m_s}{\dot{m}} \tag{39}$$

where $m_s$ is the total steel mass inside the computational domain, and $\dot{m}$ is the mass flow rate of the molten steel. In this study, $t_r = \frac{m_s}{\dot{m}} = \frac{9907 \text{ kg}}{24.15 \text{ kg/s}} = 410$ s. Thus, conventional particle simulations require significant computational times.

Moreover, some bubble sizes are rare. For example, in this work, Figure 2 shows that 0.02 mm bubbles were injected at only 900 #/s, compared with 5400 #/s 0.6 mm bubbles (2.5 million particles per residence time). Considering that there are other 9 sizes, injecting particles at every time step until reaching statistically-significant capture results is a very inefficient method.

To overcome this problem, a new methodology has been developed to find the number of bubbles with diameter $d_{pi}$ captured, $N_{ci}$, during the sample casting time, based on the corresponding number of bubbles with diameter $d_{pi}$ captured in the simulation $N_{si}$:

$$N_{ci} = \dot{R}_i \frac{l_s N_{si}}{v_c I_i} \tag{40}$$

where $\dot{R}_i$ is the injection rate (#/s) of bubbles with diameter $d_{pi}$ into a full-domain nozzle, as shown in Figure 2, $l_s$ is the measured slab sample length (m), and $I_i$ is the total number of bubbles (with diameter $d_{pi}$). These bubbles are injected as a burst, at the beginning of the DPM simulation. Then, $N_{ci}$ can be compared with the plant measurements, such as those in the UT samples of this work. This method allows any $I_i$ number of particles of a given size to be simulated and compared with measurements, so long as this number is large enough to be statistically significant. Previous work shows that at least 2500 particles are necessary to obtain accuracy within ±3% [18], and increasing the number of particles improves the accuracy. So in this work, $I_i$ = 30,000 particles were chosen for each of the 11 sizes, to achieve accuracy better than ±3%.

The overall capture fraction $C_{f,i}$ for bubbles with diameter $d_{pi}$ is defined as:

$$C_{f,i} = \frac{N_{h,i} + N_{ent,i} + N_{eng,i} + N_{o,i} + N_{r,i}}{I_i} \tag{41}$$

where $N_{h,i}$, $N_{ent,i}$, and $N_{eng,i}$ are the number of bubbles captured by the hook, entrapment, and engulfment capture mechanisms. $N_{o,i}$ and $N_{r,i}$ are the number of bubbles exiting the domain outlet and remaining in the domain at the end of the simulation.

The capture rate per meter of slab $\dot{R}_{sl,i}$ and per second $\dot{R}_{s,i}$ for bubbles with a diameter $d_{pi}$ can be calculated by:

$$\dot{R}_{sl,i} = \dot{R}_{s,i} \frac{1}{v_c} = \dot{R}_i \times C_{f,i} \frac{1}{v_c} \tag{42}$$

### 4. Casting Conditions

Two cases have been simulated on the same caster, for conditions in Table 1. The steel grade is plain low-carbon steel, typically containing 0.055% C. Argon gas is injected through the UTN and slide-gate upper plate. Only half of the domain is modeled, so the flow rates given in Table 1 are half of the real casting conditions. The two cases use 10 and 30 K superheat, which is their only difference.

**Table 1.** Caster dimensions and process conditions.

| Parameters | Symbols | Values |
|---|---|---|
| Port down angle | $\theta_s$ | 15° |
| Slab thickness (mm) | $t_s$ | 300 |
| Slab width (mm) | $w_s$ | 2300 |
| Tundish height (mm) | $h_t$ | 1400 |
| Casting speed (m/min) | $v_c$ | 0.60 |
| Steel flow rate (metric ton/min) | $\dot{m}$ | 1.45 |
| Slide-gate opening | $E/E_{max}$ | 62% |
| Submergence depth (mm) | $d_s$ | 140 |
| Argon gas flow rate | | |
| via UTN: Hot (LPM) [Cold (SLPM)] | $Q_{h\_utn}$ [$Q_{s\_utn}$] | 11.65 [2.6] |
| via plate: Hot (LPM) [Cold (SLPM)] | $Q_{h\_plate}$ [$Q_{s\_plate}$] | 4.55 [1.1] |
| Argon volume fraction | $f_{Ar}$ | 7.2% |
| Molten steel density (kg/m$^3$) | $\rho_s$ | $\rho(T) = 8278.2 - 0.7T$ |
| Molten steel viscosity (kg/m·s) | $\mu_s$ | 0.0063 |
| Liquidus temperature (K) | $T_{steel,liq}$ | 1796 |
| Steel heat capacity (J/kg·K) | $c_{ps}$ | 680 [33] |
| Steel thermal conductivity (W/m·K) | $k_s$ | 26 [33] |
| Steel thermal expansion coefficient (K$^{-1}$) | $\beta$ | $1.00 \times 10^{-4}$ [33] |
| Ar viscosity (kg/m·s) | $\mu_g$ | $2.125 \times 10^{-5}$ |
| Ar heat capacity (J/kg·K) | $c_{pg}$ | 520 |
| Ar thermal conductivity (W/m·K) | $k_g$ | 0.067 |

### 5. Flow Model Verification and Validation

Large eddy simulation (LES) is more accurate than RANS modeling because the large eddies that contain most of the turbulent energy are directly solved by LES while they are simply modeled with RANS. Here, results from the RANS model system with a ∼1.1-million-cell mesh are compared with LES of the same caster and conditions. The LES model includes a transient term in Equations (4), (8), and (15), and the Smagorinsky–Lilly model for the sub-grid-scale turbulent viscosity in Equations (8) and (15), instead of the standard k-ε turbulence model used in the RANS model. A mesh of the full domain containing ∼8 million hexahedral cells is constructed for the LES simulation, and the flow is initialized with the single-phase steady-state RANS results. Second-order upwind is used for advection terms in the LES model, and the bounded second-order implicit scheme is used for temporal discretization. It takes ∼6 months to obtain a 71 s two-way-coupled DPM-LES fluid flow with Rosin–Rammler-distributed argon bubbles injected every time step, on a 96-node high-performance computer (HPC) Mio. Details of this LES simulation can be found elsewhere [84].

The last 38 s of the LES simulation are time-averaged and compared in Figure 8 and the steady-state RANS results in the 30 K superheat. The RANS results with 10 K superheat are almost identical to those with 30 K superheat (and so are not shown).

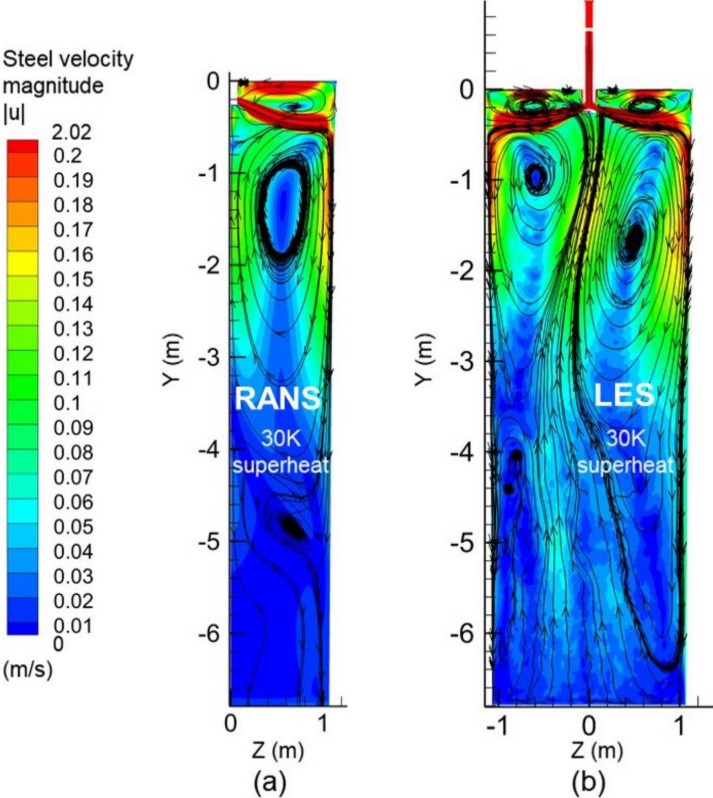

**Figure 8.** Fluid flow from (**a**) steady-state RANS simulation and (**b**) 38 s time-averaged LES simulation.

The average residence time for molten steel to flow through the strand top surface is estimated as the half-width (1.15 m) divided by the average velocity in the upper mold (0.15 m/s), which is ~8 s. Thus, 38 s should be enough to achieve an accurate time-averaged velocity near the top surface. Figure 9 compares the predicted surface velocities on a horizontal plane 10 mm below the top surface from both the RANS and the right half of LES model with nail-board measurements of the surface-flow velocity in a commercial steel plant for the same geometry and conditions.

The predicted flow is a classic double-roll flow pattern, with the flow towards the SEN in both models, which matches reasonably well with the plant measurements. The maximum velocity occurs near the width/4 point, midway between the SEN and NF. However, the velocity towards the SEN (z-component) is somewhat overpredicted by the RANS model and underpredicted by the LES model. The LES model also predicts a strong cross-flow towards the OR, which is not seen in either the RANS model or measurements. Perhaps the instantaneous surface flow pattern measurement from just one nail-board plant test in this highly-variable transient-flow region of the caster may differ from the true time-averaged behavior, even during the steady continuous casting conditions considered [85,86].

The RANS model appears to match better with the measurements than the second-order accurate LES results, perhaps because the extra numerical diffusion from the first-order upwind compensates for the extra turbulence of the jet entering the mold, as also observed in previous modeling of this flow system [87,88]. Furthermore, this first-order upwind method readily achieves a converged solution, while other RANS methods usually do not.

The LES model has other problems. The LES flow is asymmetrical in the lower strand region as shown in Figure 8b, meaning that the simulation has not reached quasi-steady state due to 71 s simulation being too short. As the average residence time in the strand, $t_r$, is 410 s, it would take ~3 years with current computational power to have a fully-developed LES flow, especially deep down the caster, which is needed for accurate particle simulations.

Thus, the new model system with the chosen RANS model is used in the current study, as the best compromise of efficiency and accuracy.

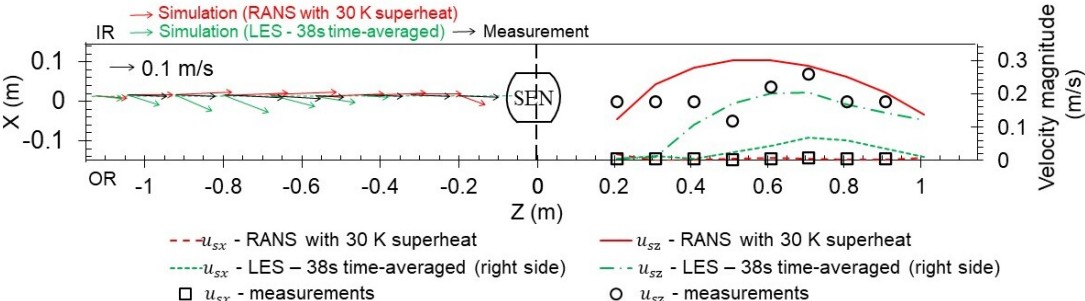

**Figure 9.** Flow model verification and validation.

## 6. Nozzle Flow Results

Figure 10 shows the molten steel velocity results and outlet argon gas distribution in the nozzle from the new model system with the RANS model. The steel mainly exits from the lower region of the nozzle port due to its higher downward momentum, as shown by the streamlines in Figure 10a. Large recirculation zones with low flow velocity are observed just below the middle plate of the slide-gate. This region has a high gas fraction, as shown in Figure 10a,b, which can cause bubble coalescence and gas pockets [55].

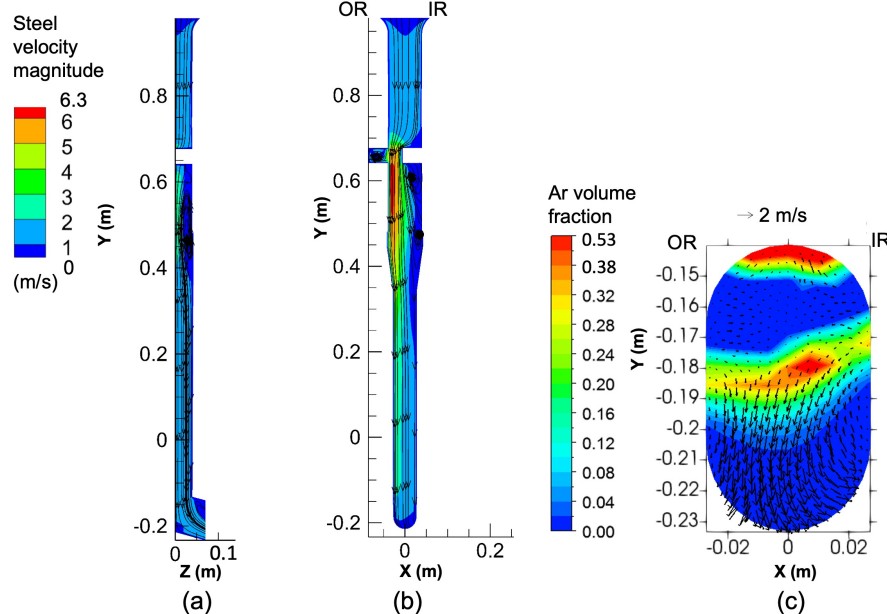

**Figure 10.** Nozzle velocity (**a**) front view; (**b**) side view; and (**c**) argon gas fraction in the plane of the outlet port.

Figure 10c shows that the steel flowing through the slide-gate opening causes a counterclockwise swirl through the left port, as shown by the arrows. This is caused by the movement of the slide-gate towards the outer radius (OR) of the caster. The argon flows out mainly from the upper region of the nozzle port, due to its buoyancy, and from the center, due to the swirling flow momentum throwing the molten steel towards the outsides of the bottom portion of the nozzle.

The jet is defined only where there is a positive outflow from the nozzle, and details of calculating its characteristics are given elsewhere [89]. Each jet has a weighted-average speed of 1.61 m/s, turbulent kinetic energy of 0.13 $m^2/s^2$, and turbulent dissipation rate

of 2.17 m$^2$/s$^3$. The jet is pointing 24.06° downward and 1.97° towards the IR face. The recirculation region where molten steel flows back into the nozzle consists of the top 39.43% of the port area.

## 7. Strand Flow Results

Figure 11 compares the strand flow patterns with 10 and 30 K superheat. The upper strand flow pattern, especially near the jet, is nearly the same for both cases as shown in Figure 11c. However, in the lower strand, increasing the superheat complicates the flow by creating a third recirculation zone as shown in Figure 11a,b. This is due to the thermal buoyancy effect and its importance can be characterized by the modified Froude number ($Fr^*$):

$$Fr^* = \frac{u_s{}^2}{gL\beta T_{super}} \tag{43}$$

where $u_s$ is the local steel velocity magnitude, and $g$ is the gravitational acceleration. $L$ is a characteristic length, taken here as the hydraulic diameter of the strand $L = \frac{2t_s w_s}{t_s + w_s} = 0.53$ m. The $Fr^*$ indicates the relative strength of inertia to buoyancy and is shown in Figure 12 for the 30 K superheat case. In the jet and upper mold region, $Fr^*$ is large, indicating that inertia is more than ten times larger than buoyancy, so thermal buoyancy is expected to have a negligible effect there. This agrees with previous work [33]. However, deeper than 3 m below the meniscus, $Fr^*$ drops to 0.1, indicating that the thermal buoyancy becomes more than ten times more important than inertia, and is expected to change the flow pattern in the the lower strand.

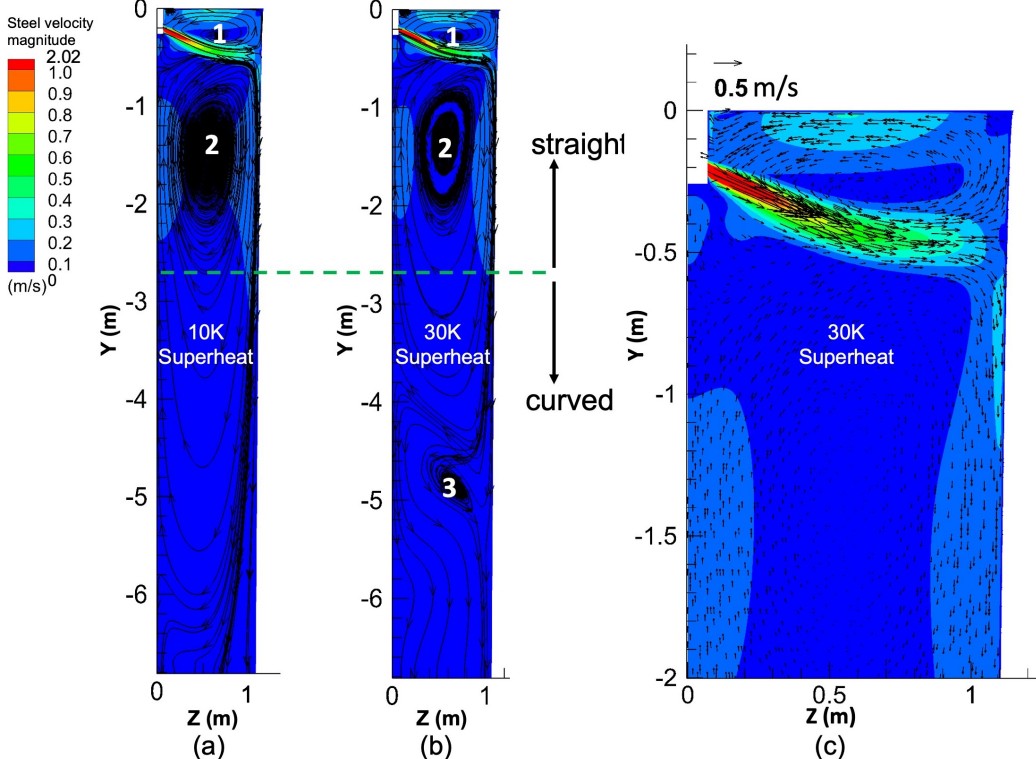

**Figure 11.** Velocity distribution in the transverse center plane: (**a**) 10 K superheat, (**b**) 30 K superheat, and (**c**) 30 K superheat close-up view.

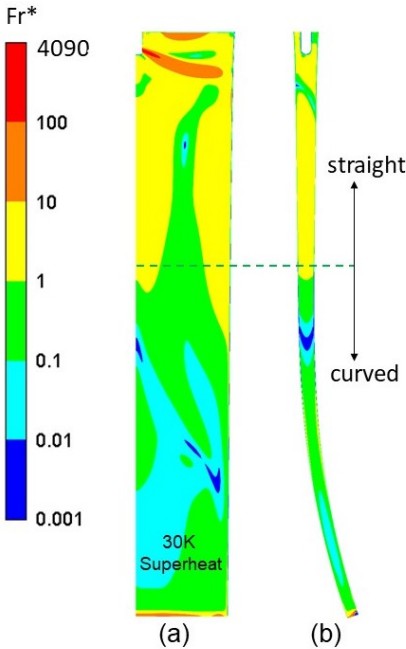

**Figure 12.** Modified Froude number distribution for 30 K superheat: (**a**) on the transverse center plane and (**b**) on the longitudinal center symmetry plane.

The internal bubble capture is related to the strand flow. Bubbles smaller than PDAS can be entrapped anywhere once they touch the solidification front. Bubbles larger than PDAS are easier to be engulfed at two places in the lower strand region: (1) in the relatively stagnant region on the IR face, and (2) in the downward-flow region near the narrow faces, where the flow velocity is close to the bubble terminal rising velocity. Particle capture bands can form in these two general regions on both the IR and OR faces.

Figure 13a–c shows the steel velocity on the width/4 plane for both superheat cases. Both results show a lower recirculation zone in the thickness/length plane with a stagnation region near the IR face. Bubbles in this region tend to stay stationary in front of the solidification front and become engulfed. Comparing Figure 13a,b, the higher (30 K) superheat shifts the stagnation region from ∼3 to ∼2.65 m below the meniscus, resulting in shallower bubble capture.

Figure 13d–f shows the steel velocity flow pattern on a thickness/length plane 0.15 m away from the NF. Figure 13d shows that the velocity down the OR face is greater with lower (10 K) superheat, so large bubbles penetrate deeper into the strand, on the OR face than IR face, where their large terminal-rising velocities are more likely to be overcome. More of these deep-penetrating bubbles are eventually captured. The terminal velocity $V_t$ can be calculated by [47]:

$$V_t = \sqrt{\frac{4(\rho_s - \rho_g)d_p g}{3\rho_s C_{D4}}} \tag{44}$$

where $\rho_g$ is the argon gas density, 0.43 kg/m$^3$. The drag coefficient $C_{D4}$ can be calculated by $\frac{24}{Re}$ for bubbles smaller than 0.9 mm [47], and the Reynolds number $Re$ is equal to $\frac{V_t d_p \rho_s}{\mu_s}$. The terminal velocity of $d_p$ = 0.6 mm bubbles is 0.218 m/s, which matches the velocity in the left top region near the OR face in Figure 13d, meaning easier capture of 0.6 mm bubbles inside that region.

Higher (30 K) superheat causes a more uniform downward flow after hitting the NF, as a result of the lower third recirculation zone (∼4–∼6 m below meniscus) pushing the velocity near the OR face up. This means that relatively more bubbles will flow downward along the IR face than OR face with higher (30 K) superheat.

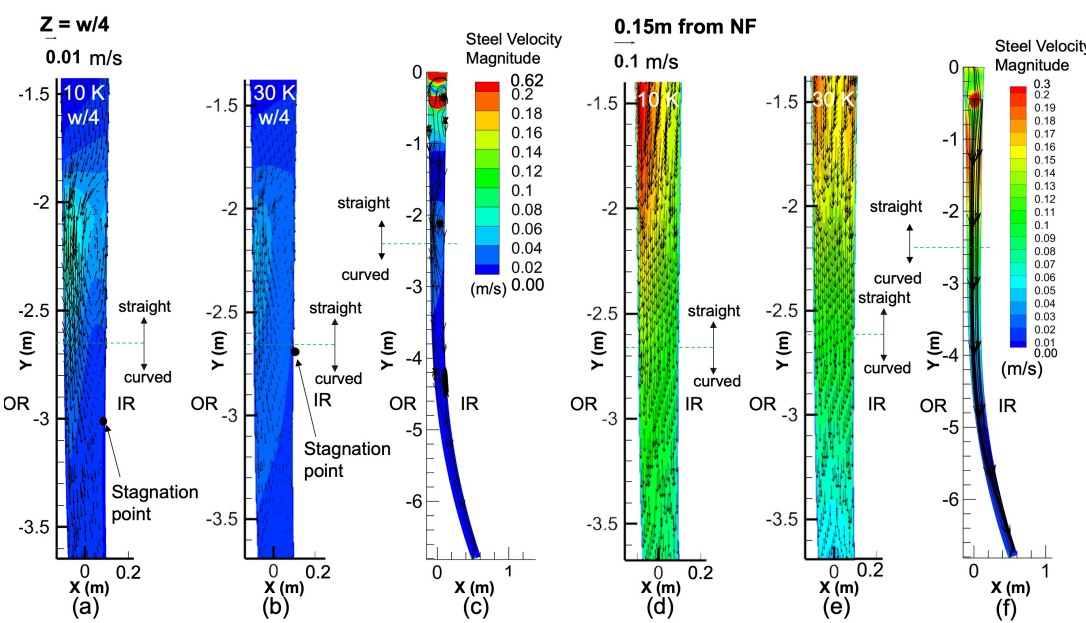

**Figure 13.** Steel velocity distribution—side view: on width/4 plane: (**a**) 10 K superheat close-up view, (**b**) 30 K superheat close-up view, and (**c**) 30 K superheat full view; on plane 0.15 m away from NF: (**d**) 10 K superheat close-up view, (**e**) 30 K superheat close-up view, and (**f**) 30 K superheat full view.

## 8. Temperature Distribution Results and Validation

Figure 14 shows the temperature distributions in the strand center plane and a horizontal plane 5 mm below the meniscus for the 10 and 30 K superheat cases. For both superheat cases, the molten steel temperature is generally higher near the jet and upper recirculation zone, and gradually cools down as the steel flows for a longer time, and is consequently transported deeper down the strand. These results mainly show the superheat temperature distribution, as the temperature contours generally have ranges of 10 and 30 K for the two cases. An important exception occurs near the top surface for the 10 K case. For this case, within 5 mm of the top surface, Figure 14c shows that the liquid temperature drops below the solidification temperature. This may form a frozen-steel crust or "island", which may become captured into the solidified shell, together with its attached slag, bubbles, and inclusions, and eventually form a giant cluster of internal defects in the final product. With 30 K superheat, the temperature everywhere is high enough to keep the steel liquid, except near the meniscus, where the two-dimensional heat losses into both the solidification front and top slag layer enable meniscus hooks to form.

In some steel samples, the UT measurements reveal a giant internal defect, which consists of a very large cluster of particles. For example, Figure 15a shows such a defect, measuring 4.9 × 587 × 1907 mm (thickness × width × length) in the plate, which corresponds to 70 × 345 × 227 mm in the slab before rolling. Plant observations show this type of defect can occur with both low (10 K) and high superheat (30 K), but is more common with a lower (10 K) superheat. This is consistent with the capture of a large frozen island/crust of the top surface, predicted in the simulations, after being broken up by rolling. Such large internal cluster defects have been observed in previous work [9,13].

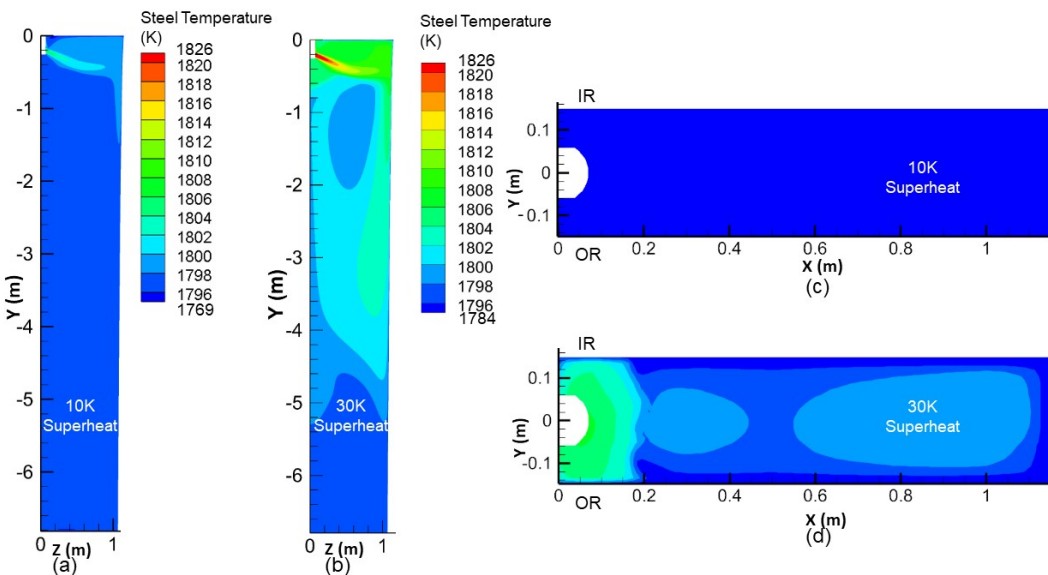

**Figure 14.** Temperature distributions: on the slab central plane front view for the (**a**) 10 K superheat and (**b**) 30 K superheat; near the top surface for the (**c**) 10 K superheat and (**d**) 30 K superheat.

## 9. Superheat Dissipation Results

For the low (10 K) superheat case, 170 kW of superheat enters the half domain through the nozzle port. The top surface slag layer is predicted to remove 57 kW (32%) of heat, as specified in Section 2.5. Much of the superheat is removed by the mold walls, including 21% (WF) and 7% (NF). The fraction removed to the mold NF is more than double its area fraction, owing to the high gradients caused by jet impingement. Below the mold, another 21% is removed from the straight strand and 17% from the curved strand. Less than 2% of the superheat exits the domain outlet by advection.

For the high (30 K) superheat case, results are generally similar, with 35%, 8%, 26%, and 20% of the 505 kW of superheat entering the half domain being removed by the mold WF, mold NF, straight strand below the mold, and curved strand, respectively. Only 0.5% of the superheat exits the domain. The top slag layers remove 57 kW heat again. However, this represents only 10% of the total superheat entering the system. This calculation is consistent with previous work [32], where only 4% of the superheat was predicted to be removed by the top surface, for a case with the 57 K superheat.

## 10. Particle Capture Model Validation

UT tests have been conducted on sixteen plates, measuring $0.21 \times 3.9 \times 36.2$ m (thickness $\times$ width $\times$ length) rolled from $0.3 \times 2.3 \times 4.31$ m slabs samples. Figure 15 shows an example of the UT tests for a case with 29 K superheat.

Figure 15a shows an end view of the measured particle capture locations, converted back to the original slab dimensions. The black dots are extracted from the UT map by an online tool WebPlotDigitizer. Most of the captured particles are located around the transition line from vertical to curved (the green dashed line) on the IR face, forming a capture band (between the two red dashed lines), especially on the inside radius. This capture band is due to the stagnation in this region shown in Figure 13b, as discussed previously. It is located 40–60 mm below the strand surface, which corresponds to ~1.6–3.5 m below the meniscus along the strand, as shown in Figure 15b, This is similar to the location of capture bands observed below the surface in previous work: 50–65 mm in [21] and 35–45 mm in [22].

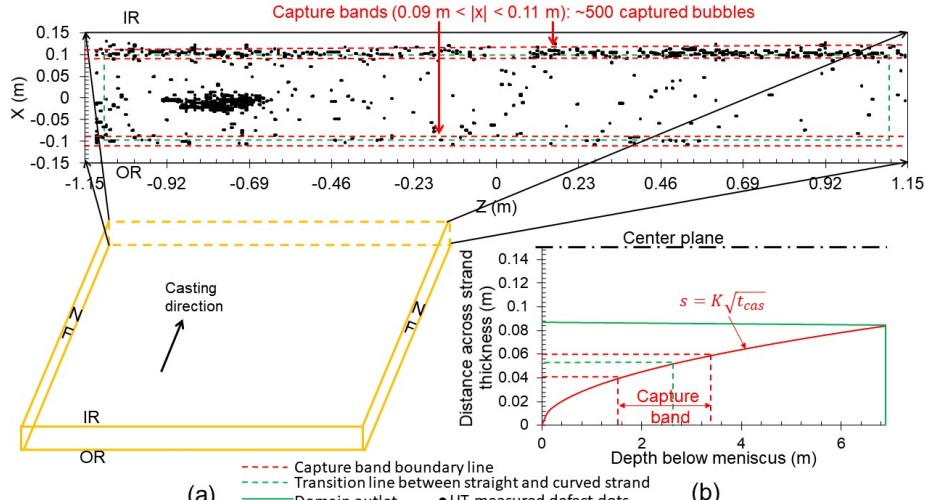

**Figure 15.** UT measurements: (**a**) Schematic showing slab and particle capture locations; (**b**) capture band location on the shell thickness profile.

Figure 16 shows results for a different slab. The red dots show the calculated capture locations of 0.6 mm bubbles with 10 K (Figure 16a) and 30 K superheat (Figure 16b). The latter results are compared with another UT map (black dots), and reveal 202 particles in the capture band. Only the capture locations of large ($d_p > 0.5$ mm) bubbles are compared with the UT results because the latter cannot detect any small defects. Near the meniscus, the overprediction is likely caused by the difficulty in detecting the internal particles near the plate surface by the UT device. The simulation predicts more bubbles captured with a lower (10 K) superheat, especially inside the IR and OR capture-band regions (between the upper and lower pair of red dashed lines), due to the deeper stagnation region near the IR face and higher downward velocity near the OR face.

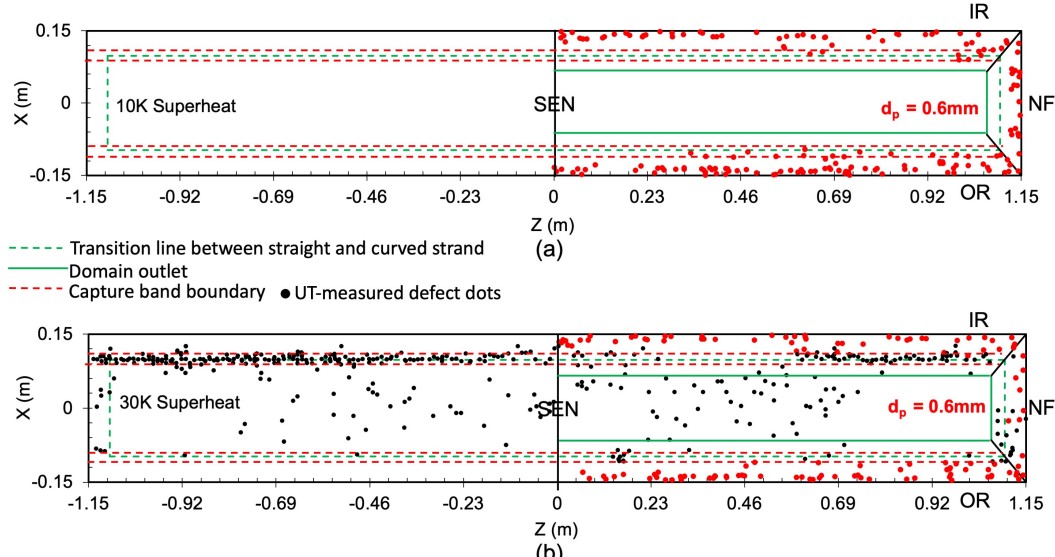

**Figure 16.** Capture locations of 0.6 mm bubbles comparing simulations and UT measurements—end view—(**a**) 10 K superheat and (**b**) 30 K superheat.

In Figure 16b, far fewer bubbles are predicted in the capture-band region than measured. This is because only 30,000 of the 0.6 mm bubbles are injected—far less than the actual number of 0.6 mm bubbles in the strand during the same sample casting time, as explained in Section 3.

Equation (40) can be used to estimate the true number of bubbles captured inside the capture band of the sample, $N_{ci}$, by setting $i = 0.6$ mm. These post-processed results for both 10 and 30 K simulations are compared with the 16 UT measurements of rolled slab samples in Figure 17. The predictions are now quantitatively consistent with the measurements. The higher superheat case shows fewer defects in both IR and OR, which agrees with the plant measurements. Both the simulation and measurements show that the number of bubbles captured inside the capture band gradually decreases with increasing superheat. With higher superheat, more bubbles are captured on the IR face than on the OR face. With lower superheat, the trend is reversed as more bubbles are captured on the OR face. These results, which are observed in both simulation and measurements, can be explained by the flow pattern in Figure 13, as discussed previously.

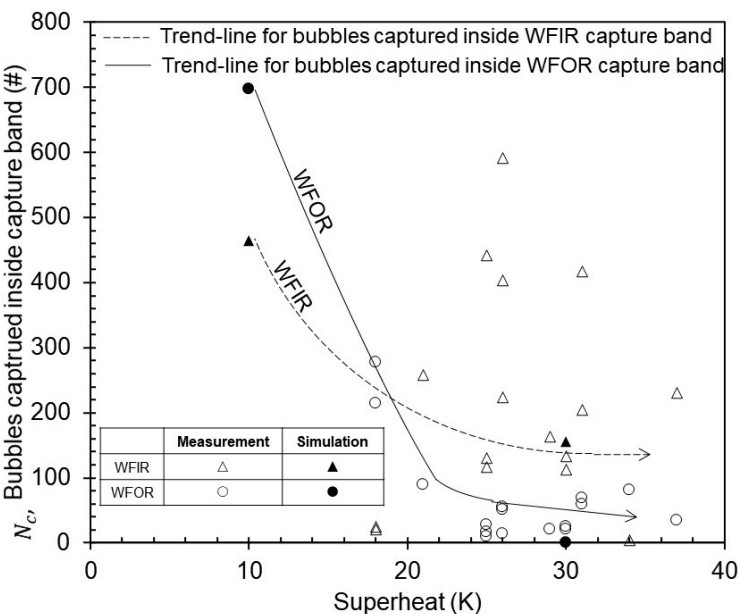

**Figure 17.** Model validation—number of bubbles captured in the capture band comparing simulation and UT measurements.

## 11. Particle Capture Results

Figure 18a–d shows an end view of capture locations for 0.1 and 0.3 mm bubbles with 10 and 30 K superheat. In addition to many particles captured near the strand surface, the model predicts a capture band on both the IR and OR faces, near the transition lines between the straight and curved parts of the strand. These capture bands are due to bubbles being entrapped while moving down near the NF into the lower recirculation zone. This leaves a thin, relatively clean space between the surface region and the capture band.

Figure 18e–h shows where 0.1 and 3 mm bubbles escape into the top slag layer for both superheat cases. Small bubbles ($d_p < 0.5$ mm) tend to float up and escape uniformly over the top surface area. Large bubbles ($d_p > 0.5$ mm) tend to escape near the SEN, due to their large buoyancy floating them up immediately after exiting the nozzle port.

Figure 19 shows the capture locations of different bubbles on the wide-face inner radius (WFIR) for both superheat cases. For both cases, smaller bubbles ($d_p < 0.5$ mm) flow deeper down the strand and are mostly captured, because they tend to flow with the molten steel and be transported deeper. Large bubbles ($d_p > 0.5$ mm) tend to flow up into the slag layer, and very few of them are captured. For 0.3 mm bubbles, a sudden increase of capture occurs at the distance down the caster where the PDAS size increases to the bubble diameter. With a higher (30 K) superheat, bubble capture locations are shallower due to larger thermal buoyancy, which lessens particle penetration depth. The results here suggest

that 6 and 2 bubbles are captured inside the measured capture-band region for 10 and 30 K superheat cases, respectively.

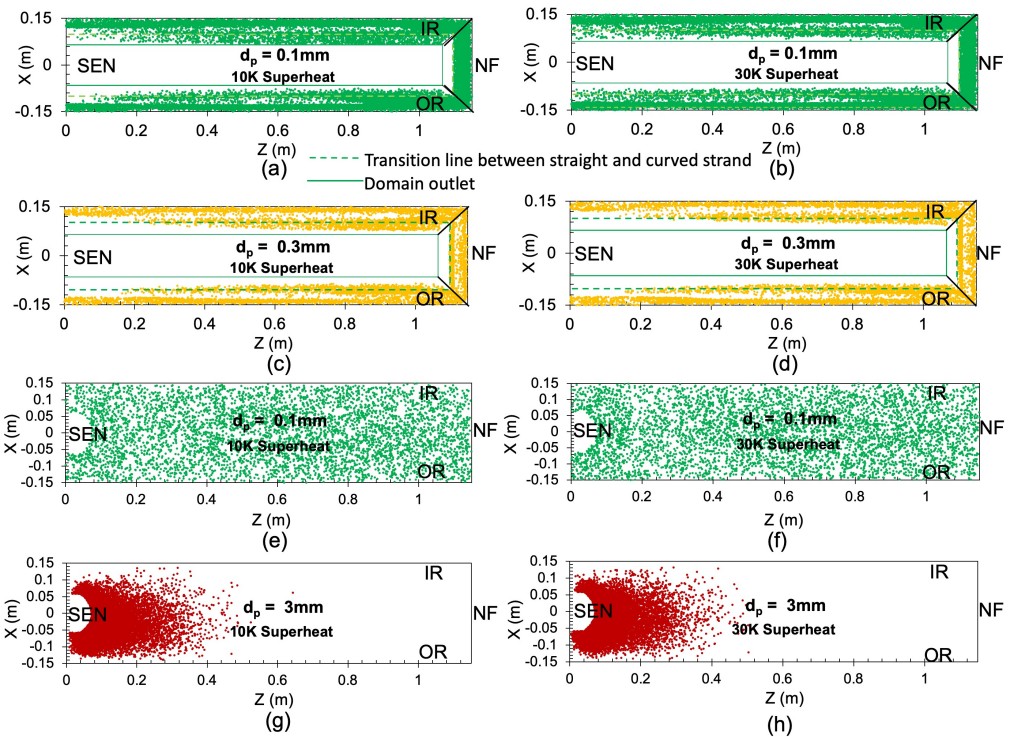

**Figure 18.** Capture and removal locations of selected bubble sizes: (**a**–**d**) capture endview; (**e**–**h**) removal top view; (**a**,**c**,**e**,**g**) 10 K superheat; (**b**,**d**,**f**,**h**) 30 K superheat.

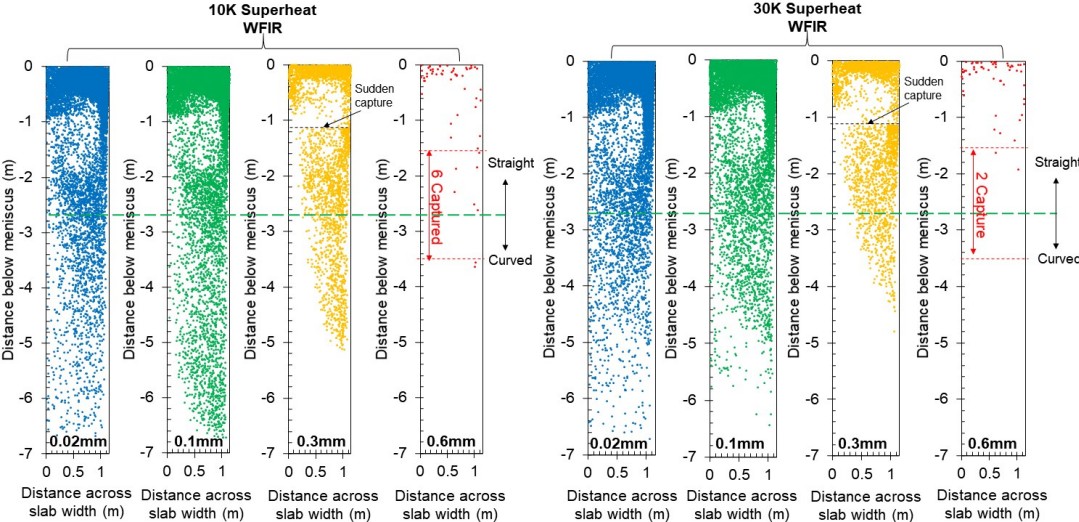

**Figure 19.** Capture locations on the wide face inner radius (WFIR) for selected bubble sizes—front view.

Figure 20 shows the capture locations of different bubble sizes on the wide-face outer radius (WFOR) for both superheat cases. Trends are similar to the IR in Figure 19. The lower (10 K) superheat case leads to more capture of 0.6 mm bubbles on the OR face inside the measured capture band, while the higher (30 K) superheat case shows the opposite trend. This is due to the flow pattern differences, shown in Figure 13d. The higher downward velocity near the OR face with lower (10 K) superheat transports more 0.6 mm bubbles deeper down the face and balances their terminal velocities.

Figure 21 shows the capture locations of different bubble sizes on the NF for both superheat cases. Higher superheat (30 K) exhibits shallower capture for all bubble sizes.

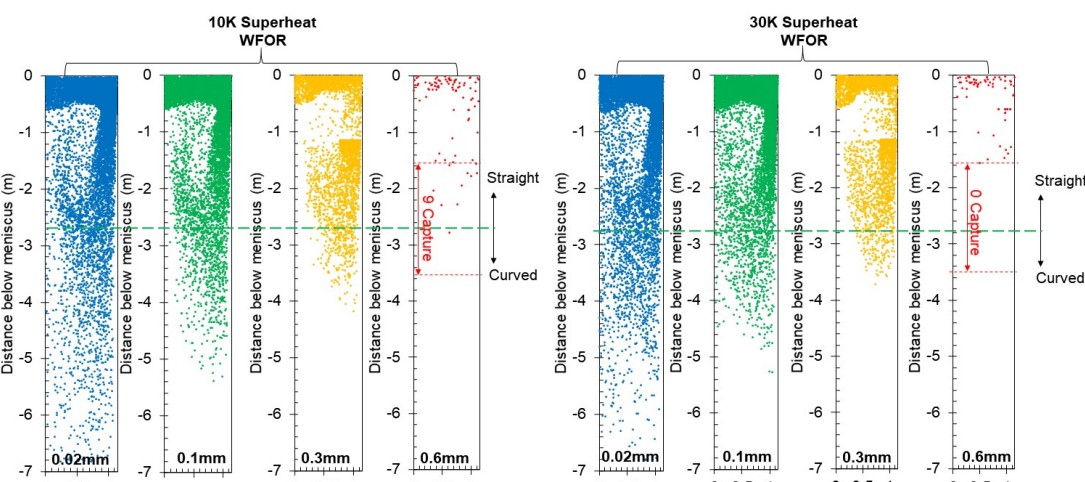

**Figure 20.** Capture locations on the wide face outer radius (WFOR) for selected bubble sizes—front view.

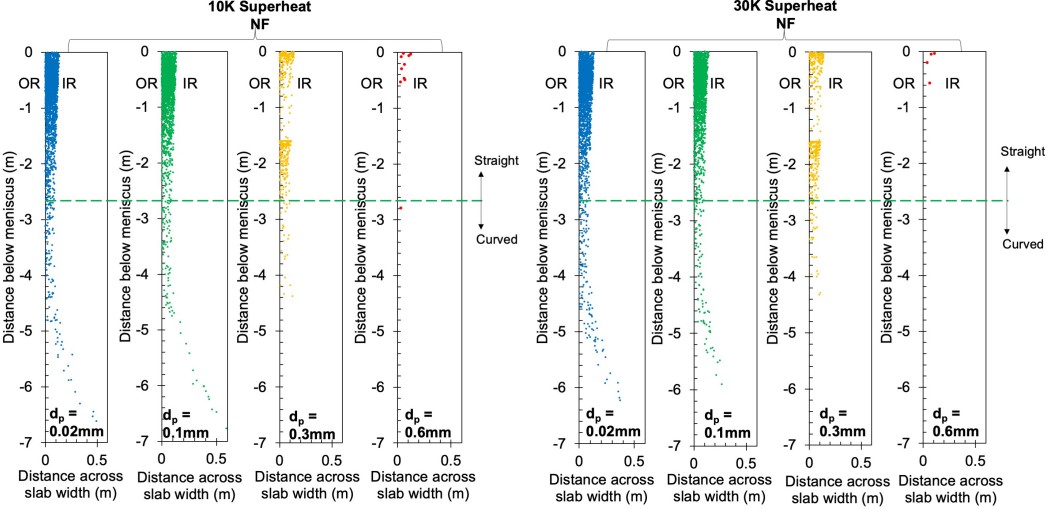

**Figure 21.** Capture locations on the narrow face (NF) of the selected bubble sizes—side view.

Figure 22 shows the capture and escape fractions by different regions of the strand for both superheat cases. The escape fraction into the top slag layer increases with increasing bubble diameter. Only ∼9% of 0.02 mm bubbles are removed by the slag layer, while almost all 0.6 mm bubbles float up and escape into the top slag layer. Higher (30 K) superheat has a slightly higher escape fraction.

In the hook zone, the lower (10 K) superheat case has higher capture by all three mechanisms (meniscus hooks, entrapment, and engulfment) relative to the 30 K case. For small bubbles ($d_p < 0.5$ mm), hook capture is negligible compared with the entrapment and engulfment mechanisms. The hook capture mechanism becomes increasingly important with increasing bubble size. For very small bubbles, ($d_p < 0.06$ mm), entrapment is the dominant capture mechanism everywhere, including inside the hook zone. Engulfment grows in importance with increasing bubble size. For all bubble sizes, capture in the hook zone is greater with the lower (10 K) superheat case. Several (16) large bubbles ($d_p > 0.5$ mm) are captured by the hook mechanism with lower (10 K) superheat, due to the deeper meniscus hooks. This includes a single 1 mm bubble. With the high superheat case, no large bubbles ($d_p > 0.5$ mm) are captured in the hook zone.

In the straight strand, including the mold, over 70% of 0.02 and 0.1 mm bubbles are captured by the shell (entrapment/engulfment). Only ~10% of these small bubbles are captured by the curved part. For 0.6 mm bubbles, ~0.016% are captured by the curved part with the 10 K superheat, but no large bubbles ($d_p > 0.5$ mm) are captured in this region with the higher (30 K) superheat. As expected, and observed in the plant, more bubbles are captured on the IR in the curved part of the strand.

Figure 23 compares the overall capture fraction of different bubbles with 10 and 30 K superheat using Equation (41), and their capture fractions by the hook, entrapment, and engulfment mechanisms.

For small bubbles ($d_p < 0.5$ mm), the overall capture fraction is very similar for both superheat cases. For $d_p < 0.1$ mm bubbles, the overall capture fraction reaches ~90% and gradually declines with increasing bubble sizes. For large bubbles ($d_p > 0.5$ mm), the higher (30 K) superheat case has slightly less overall capture (0.05% vs. 0.06%). The single bubble captured that was larger than 1 mm was by the hook capture mechanism with 10 K superheat, and represents 0.003% of those injected.

All of the other $d_p > 1$ mm bubbles escape into the top slag layer for both superheat cases. This agrees with previous work that almost all large bubbles escape to the top surface [16,20,24]. For both superheat cases, entrapment dominates over engulfment for $d_p < 0.3$ mm bubbles, because these bubbles are smaller than PDAS and can easily flow in between two primary dendrite arms to be captured whenever they touch the solidification front. For $d_p > 0.3$ mm bubbles, engulfment naturally becomes more important.

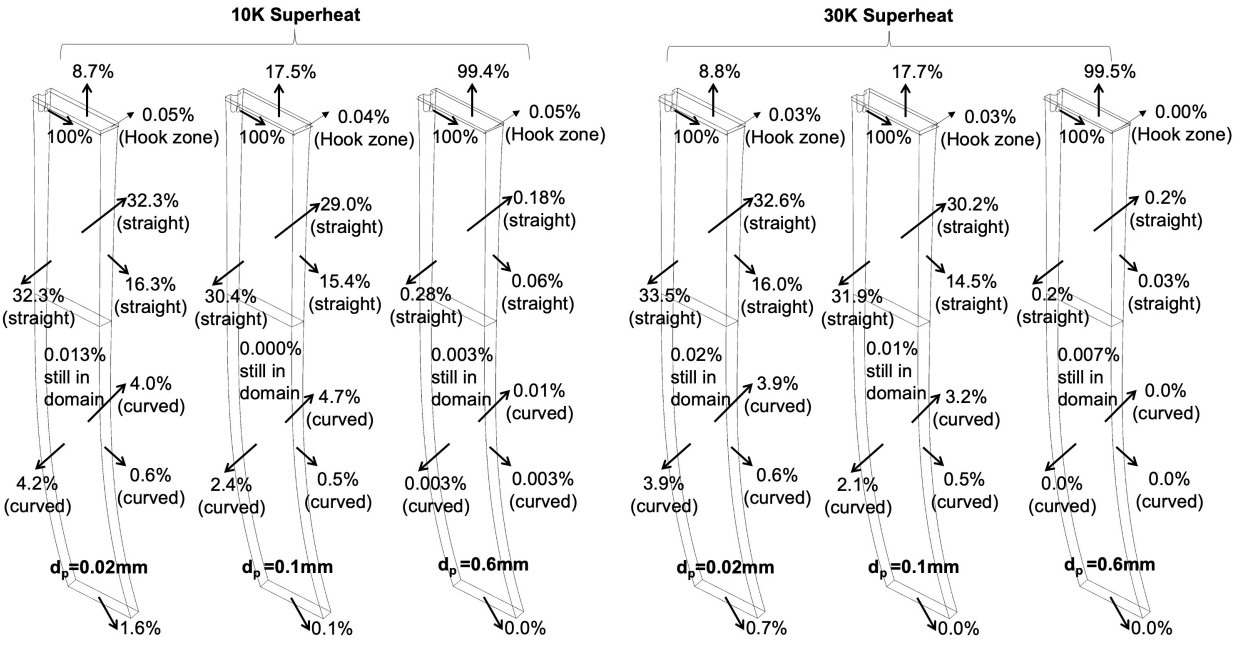

**Figure 22.** Capture/escape fractions to different strand regions for selected bubble sizes.

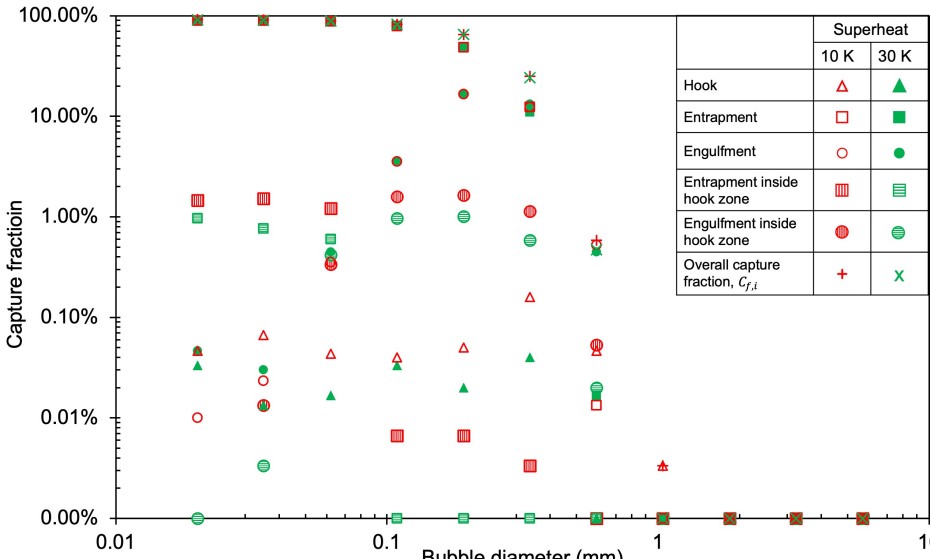

**Figure 23.** Overall capture fraction of different bubbles, and breakdown by hook, entrapment, and engulfment mechanisms.

Figure 24 compares the capture rates in the slab samples for the 10 and 30 K superheat cases using Equation (42), where the overall capture fraction of bubbles with diameter $d_{pi}$, $C_{f,i}$, is shown in Figure 23.

There are about eleven 1 mm bubbles captured per meter of the slab with 10 K superheat, while no bubbles larger than 1 mm are captured with higher (30 K) superheat.

The new findings presented here show the important drawbacks of low superheat on meniscus solidification and fluid flow deep in the caster, and the corresponding effects on increasing particle-capture defects. This work neglects the important effects that the superheat has on fluid flow and the inclusion of particle distribution in the tundish. These differences in tundish flow behavior serve to greatly augment the effects of superheat in the strand, which are identified here.

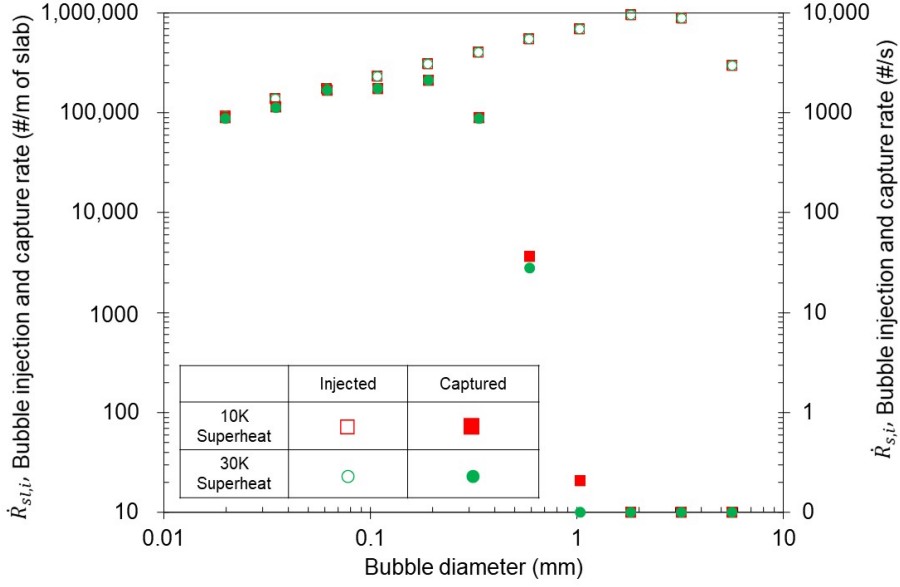

**Figure 24.** Injection and capture rates for all bubble sizes.

## 12. Conclusions

A new modeling approach was developed to investigate the multiphase turbulent flow, superheat dissipation, particle transport, and capture during steady continuous casting of steel slabs. The predictions were validated with plant measurements including nail board dipping tests of surface velocity and UT measurements of particle capture locations. Good numerical convergence (to within $10^{-4}$ for the nozzle flow and $10^{-7}$ for the strand flow) was achieved using a half nozzle and strand domain with structured hexahedral cells aligned with the jet direction.

Findings regarding the fluid flow include:

- Superheat has a negligible effect on the flow in the upper strand region where inertia is dominant over thermal buoyancy.
- Superheat has a significant effect in the lower strand, especially >3 m below the meniscus where thermal buoyancy dominates the flow and creates multiple complicated recirculation regions.
- The flow pattern shows two locations for large bubbles to be engulfed: (1) near the stagnation point on the IR face, where the bubble tends to stay stationary, and (2) in the downward flow region near IR and OR faces, where the downward velocity balances the bubble terminal velocity. Capture bands can form in these two locations, which agrees with plant measurements.
- The lower (10 K) superheat case has a deeper stagnation region near the IR face and higher downward flow near OR face due to less thermal buoyancy, which allows steel to flow deeper and faster down the narrow faces below the jet impingement, leading to more and deeper capture.
- The lower (10 K) superheat case appears susceptible to a possible frozen crust or island near the top surface due to lower inlet temperature and heat loss into the slag layer. The capture of this island may bring slag, bubbles, and inclusions deep into the strand, forming a giant internal defect in the final product.
- The higher (30 K) superheat case leads to a more uniform downward velocity and a shallower stagnation region near the IR face, due to the lower third recirculation zone pushing the flow upwards.

The bubble transport and the capture results show:

- With the lower (10 K) superheat case, more 0.6 mm bubbles are captured on the OR face in the capture-band region than the IR face, due to the higher downward velocity transporting the bubbles deeper and balancing out their terminal velocities (0.218 m/s).
- The lower (10 K) superheat case leads to deeper meniscus hooks, which capture more particles, leading to more surface defects.
- The higher (30 K) superheat case leads to fewer and shallower capture of all bubble sizes as a result of stronger thermal buoyancy, indicating fewe internal defects.
- The higher (30 K) superheat case leads to relatively more 0.6 mm bubbles captured on the IR face than OR face in the capture-band region, due to a more uniform downward flow.
- Capture bands are predicted at ∼1/4 thickness from both the IR and OR strand surfaces, which matches UT maps.
- For small bubbles ($d_p < 0.5$ mm), clear capture bands are seen on both IR and OR faces near the transition line from vertical to curved parts of the strand from the end view. These two capture bands are due to bubbles being entrapped while moving down near the NF into the lower recirculation zone. This leaves a clean space.
- Small bubbles ($d_p < 0.5$ mm) tend to escape into the slag layer uniformly over its surface, while large bubbles ($d_p > 0.5$ mm) escape near the SEN due to the larger buoyancy, bringing them up immediately after exiting the nozzle port.

- The escape fraction into the top slag layer increases with increasing the bubble size due to the increasing buoyancy effect ($\sim$9% to $\sim$100% for 0.02 and 0.6 mm bubbles, respectively).
- Superheat has little effect on the capture of bubbles smaller than 0.5 mm because they tend to flow with the steel, move down deeply, and become entrapped.
- Increasing superheat decreases the capture of 0.6 mm bubbles by 20% due to the complicated recirculation regions hindering the penetration of large bubbles deep into the caster
- Almost all very large bubbles ($d_p > 1$ mm) escaped into the top slag layer due to large buoyancy.
- For $d_p < 0.3$ mm bubbles, entrapment dominates over engulfment, because the bubbles are smaller than PDAS. As the bubble size increases $> 0.3$ mm, engulfment becomes increasingly important.
- Inside the hook zone: For small bubbles ($d_p < 0.5$ mm), the hook capture is negligible, relative to the entrapment and engulfment mechanisms. For very small bubbles ($d_p < 0.06$ mm), entrapment is the dominant mechanism, although engulfment grows in importance with the increasing bubble size. The single captured 1 mm bubble is by the hook capture mechanism with 10 K superheat.

**Author Contributions:** Conceptualization, methodology, software, formal analysis, investigation, validation, visualization, writing—original draft preparation, M.L.; methodology, software, supervision, S.-M.C.; Data curation, resources, investigation, X.R.; conceptualization, methodology, formal analysis, investigation, resources, supervision, writing—review and editing, project administration, funding acquisition, B.G.T. All authors have read and agreed to the published version of the manuscript.

**Funding:** This research was funded by the National Science Foundation (grant no. CMMI 18-08731), the Continuous Casting Center, Colorado School of Mines, and Baosteel, Shanghai, P.R. China (grant no. 470108).

**Acknowledgments:** The authors thank Baosteel, Shanghai, P.R. China, for providing the casting conditions and plant measurements. Provision of Fluent licenses through the Ansys Inc. academic partnership program is much appreciated.

**Conflicts of Interest:** The authors declare no conflict of interest.

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
