# Peer review of "Modeling of Multiphase Flow, Superheat Dissipation, Particle Transport, and Capture in a Vertical and Bending Continuous Caster"

_processes, doi:10.3390/pr10071429_

Round 1

Reviewer 1 Report

In this study, a new particle entrapping model in steel continuous casting process is proposed, which includes the effects of multiphase flow injected by argon and thermal buoyancy generated by overheating of steel strands. The capture of solidification hooks at the meniscus, the capture between dendrites and the engulfment around large particles were simulated.The new model system is of great significance for bubble prediction. 

L172, The argon flows out mainly from the center and upper region of the nozzle port, according Figure 8a nozzle velocity is towards the center of the port, which is different from the figure c, explain the difference. 

L696, With higher (30 K) superheat, bubble capture locations are shallower due to larger thermal buoyancy, which lessens particle penetration depth. For particles with small particle size, there seems to be little difference. Is the effect of superheat selective to size? 

The superheat cases are lower (10 K) and higher (30 K), Why choose these two conditions? Because in many cases, the superheat degree in the mold is less than 30 degrees. 

Figure 3. Shell thickness profile: (a) On wide face (WF),Wide face Shell, the letter S in shell should be lowercase letter.

Author Response

The authors wish to thank the reviewers for their great comments, questions, and suggestions, and we have made many changes to our manuscript to address them. A point-by-point reply to each question follows.

Response to Reviewer 1 Comments

In this study, a new particle entrapping model in steel continuous casting process is proposed, which includes the effects of multiphase flow injected by argon and thermal buoyancy generated by overheating of steel strands. The capture of solidification hooks at the meniscus, the capture between dendrites and the engulfment around large particles were simulated. The new model system is of great significance for bubble prediction. 

Thank you.

Point 1: L172, The argon flows out mainly from the center and upper region of the nozzle port, according Figure 8a nozzle velocity is towards the center of the port, which is different from the figure c, explain the difference. 

Response 1: Thank you for this good question. The streamlines and contour in figure 8a represent the steel velocity, not the argon. The steel flows mainly from the center and lower regions of the nozzle outlet. Figure 8c shows the steel velocity by the arrows and the argon volume fraction by the contours at the nozzle outlet. The argon gas mainly exits the port at the center and upper region due to its buoyancy. We revised the text to make this more clear for the reader - please see lines 599 – 610 of the revised paper.

Point 2: L696, With higher (30 K) superheat, bubble capture locations are shallower due to larger thermal buoyancy, which lessens particle penetration depth. For particles with small particle size, there seems to be little difference. Is the effect of superheat selective to size? 

Response 2:

Yes: superheat is selective to size. You are reading our results correctly.

The effect of superheat is stronger on the large particles than on the small particles. This is because the small particles travel with the flow down the caster, as their buoyancy is very small. So, the small particles (dp < 0.5 mm) flow with the steel and are captured deep in the strand for both high and low superheat, so are not much affected by superheat, as you correctly observe. Thus, their capture fractions are very close as seen in Figure 23.

On the other hand, the capture of large particles is greatly affected by the superheat, owing to their great buoyancy, so almost all of them escape into the top slag layer and very few of them are captured. The multi-recirculation zones caused by higher superheat can prevent their penetration downwards and reduce their capture with higher superheat.

Point 3: The superheat cases are lower (10 K) and higher (30 K), Why choose these two conditions? Because in many cases, the superheat degree in the mold is less than 30 degrees. 

Response 3: We chose these two cases based on real-plant observations (Baosteel). The superheats used in the plant range from less than 10 K to over 30 K superheat (sometimes even 37 K), so these two superheats are typical examples of “high” and “low” superheat at the plant. Plant observations found that the 30 K superheat gives better steel quality then 10 K superheat, so these two conditions were chosen for careful analysis and evaluation.

Point 4: Figure 3. Shell thickness profile: (a) On wide face (WF),Wide face Shell, the letter S in shell should be lowercase letter.

Response 4: Thank you for spotting the typo, and it has been corrected.

Reviewer 2 Report

It is a well written article that involves a modeling of multiphase flow encountered in the process of continuous casting of steel. Following are some points that require further explanation

1) Why is an Eulerian-Eulerian approach adopted to model the molten-steel and argon-gas mixture when the argon-gas bubbles are also simulated in a Lagrangian framework? Argon-gas bubbles simulated using a pure DPM approach and a single-phase Eulerian approach for molten steel probably should be sufficient to capture important physics studied here.

2) Is the domain on which the EE-multiphase flow solved, changing throughout the simulation (due to the solidification process) or is it defined at the beginning of the simulation?

3) In the EE approach, why is the energy equation not being solved for the argon-gas? There should be some energy exchange between the two phases.

4) No grid convergence study is performed. How did you arrive to the mesh resolution used in this study?

5) No information is provided on the numerical methods being used. Example, linear solvers, pressure-velocity coupling algorithm, etc.

6) Equation 40 is valid only if particle-particle interactions are being neglected in the Lagrangian framework. This assumption needs to be explicitly stated. Also, what kind of impact on results do you envision if bubble-coalescence and breakup were considered.

7) Line 417; instead of switching to upwind to maintain stability did you consider using TVD like schemes for gradient computation. First order upwind can lead to very diffusive results if mesh resolution is not fine enough.

8) Line 161; why is a 2-D heat conduction equation being solved for the mold wall? If the mold wall has a finite thickness and transverse gradients are important, a 3-D heat equation should have been solved.

9) Please make sure to define the abbreviations first time they are used. Also, including them in figure captions makes it easier to understand the results.

Author Response

The authors wish to thank the reviewers for their great comments, questions, and suggestions, and we have made many changes to our manuscript to address them. A point-by-point reply to each question follows.

Response to Reviewer 2 Comments

It is a well written article that involves a modeling of multiphase flow encountered in the process of continuous casting of steel. Following are some points that require further explanation

Thank you for your kind comment and great questions.

Point 1: Why is an Eulerian-Eulerian approach adopted to model the molten-steel and argon-gas mixture when the argon-gas bubbles are also simulated in a Lagrangian framework? Argon-gas bubbles simulated using a pure DPM approach and a single-phase Eulerian approach for molten steel probably should be sufficient to capture important physics studied here.

Response 1: Thank you. We have made several revisions to the paper to better explain and justify our approach, and to answer these questions for the reader. The purpose of using Eulerian-Eulerian approach is to efficiently obtain a converged solution, to obtain an accurate two-way-coupled steady-state flow model, together with an accurate calculation of the transport and potential capture of many (330,000) particles. The argon gas has a very important effect on the fluid flow, owing to its buoyancy pushing the steel jets upwards, so this effect must be included in the model. A two-way-coupled simulation using a pure DPM approach for the argon bubbles and single-phase Eulerian approach for molten steel is another way to simulate these phenomena, and we have used this approach in previous research. However, the latter approach takes considerable amount of time to reach quasi-steady state, because it requires the DPM particles to properly accumulate in the domain. In the system of this paper, the residence time of the molten steel in this 7-m-long strand is 410 s, which would take many months to simulate even with 96 nodes on the HPC (high performance computer) Mio of Colorado School of Mines. Moreover, the number of large particles entering the domain during this time would still be statistically too small in the lower strand region, so many more months of simulation would be required. The Eulerian-Eulerian approach, on the other hand, is able to give results within a reasonable time frame, while capturing the important physics just as well. Having the fluid flow pattern, any number of DPM particles can be injected, tracked, and captured, as a very efficient, uncoupled calculation.

Point 2: Is the domain on which the EE-multiphase flow solved, changing throughout the simulation (due to the solidification process) or is it defined at the beginning of the simulation?

Response 2: You are correct that the domain is defined at the beginning of the simulation and never changes throughout the simulation. The solidification process is accounted for by first calculating the shell thickness using the empirical equation in section 2.2 and using the results, to create a domain of only the liquid pool of the strand, without the solidified shell. This has many advantages, including faster convergence, no need for extremely small cells to resolve the solidification front, and the use of accurate wall functions at the solidification front. Changes to the section 2.9 were made to add this point. Another huge benefit is the natural thermal boundary condition of the liquidus temperature at the solidification front, which avoids the need to include the mold, interfacial gap, and complicated spray cooling boundary conditions in the FLUENT model (these complicated phenomena are only needed in the CON1D model, which matches with the empirical measurements of the shell thickness). 

A sentence was added to summarize the above into the text in lines 423 - 425.

Point 3: In the EE approach, why is the energy equation not being solved for the argon-gas? There should be some energy exchange between the two phases.

Response 3: The energy equation actually is being solved for both the molten steel and argon gas, owing to this feature being included in Fluent automatically. However, in reality, the argon bubbles, as individual particles, are always moving with the molten steel at the same temperature of the local molten steel surroundings, The heat carried by the argon bubbles is negligible due to its very small mass fraction (0.0044%) and also due to its very low heat capacity, density, and thermal conductivity. This means that almost all the superheat is conveyed by the molten steel, and the argon gas bubbles have a negligible effect. To avoid making the reader interested in this negligible phenomenon, and to save space in the paper, the energy equation for the argon bubbles is not shown.

Point 4: No grid convergence study is performed. How did you arrive to the mesh resolution used in this study?

Response 4: First, the grid size is comparable with previous work. For instance, a recent work from Seong-Mook Cho, et al. (Cho, S.-M.; Thomas, B.G.; Hwang, J.-Y.; Bang, J.-G.; Bae, I.-S. Modeling of Inclusion Capture in a Steel Slab Caster with Vertical Section and Bending. Metals 2021, 11, 654. https://doi.org/10.3390/met11040654) run a similar simulation on a 9.5-m-long full-domain slab and gets good validation with plant measurements, with 1.64 million cells. Therefore, mesh sensitivity study is not performed in this paper, and 1.1 million cells are considered as statistically enough for a half-domain nozzle and strand. As a matter of fact, too many cells will cause divergence of the solution, due to large numerical error. The author’s previous work shows (not published) that ~4 million cell on the same 7-m-long full-domain nozzle and strand leads to diverge (oscillation results). Plus, we did a time-averaged LES simulation with a ~8-million-cell mesh (will be published soon in Mingyi Liang’s Ph.D. thesis) as a verification of our RANS results with this ~1.1-million-cell mesh, and they match okay. In addition, our RANS model results match better with the measurements than the LES results.

Please see the new section 5 for the added comparison between the RANs and LES results.

Point 5: No information is provided on the numerical methods being used. Example, linear solvers, pressure-velocity coupling algorithm, etc.

Response 5: Thank you for catching this oversight. We have revised the text to include this and other numerical details: a Gauss-Seidel linear equation solver and an Algebraic Multigrid (AMG) method in ANSYS FLUENT with its Coupled algorithm, on a staggered grid with zero velocities everywhere as initial conditions. This algorithm simultaneously solves the equations of velocity corrections of each phase and pressure correction and is very efficient in steady-state simulations.

Please see line 496-498 for the added explanation of the linear solvers and pressure-velocity coupling algorithm.

Point 6: Equation 40 is valid only if particle-particle interactions are being neglected in the Lagrangian framework. This assumption needs to be explicitly stated. Also, what kind of impact on results do you envision if bubble-coalescence and breakup were considered.

Response 6: Thank you for your great insight.

Yes, equation 40 is valid only if particle-particle interactions are being neglected in the Lagrangian framework. The neglect of particle-particle interactions is explained in the DPM model section (lines 310-312).

Instead of modeling coalescence and breakup, as done by others, a Rosin-Rammler distribution is assumed for the injected argon bubbles. Coalescence and breakup are important phenomena in the nozzle region where the gas is confined with a high volume fraction. This region is handled by imposing a Rosin-Rammler size distribution, which represents the bubble size distribution after coalescence and breakup at the nozzle outlet and is injected at the gas inlet. After the gas bubbles enter the mold region, the bubble size distribution varies due to their different residence times. However, breakup is negligible, because the turbulence is so much lower, and coalescence is negligible due to the low gas fraction and rare bubble-bubble interactions. In addition, surface tension acts to keep most of the bubbles spherical.

The above text is added to include these points in lines 151 – 158.

Point 7: Line 417; instead of switching to upwind to maintain stability did you consider using TVD like schemes for gradient computation. First order upwind can lead to very diffusive results if mesh resolution is not fine enough.

Response 7: No, we did not use TVD, but it will be considered in future work. Thanks for pointing this out, as TVD is very efficient in combining first and second order for stability.

Yes, first order upwinding naturally leads to extra diffusion, especially if the mesh resolution is not fine or the mesh has an angle with the flow direction. We make the mesh to be as aligned to the flow direction in both nozzle and strand as possible, so the numerical diffusion is lessened.

To properly answer this question, the validation section 5 of our paper is expanded to compare with a Large Eddy Simulation (LES) model that we ran but did not previously include.

In section 5, the first-order RANS model is compared with 2nd-order LES model. The RANS appears to match better with the measurements than the second-order accurate LES results, perhaps because the extra numerical diffusion from the first-order upwinding compensates for the extra turbulence of the jet entering the mold, which is also seen in previous work. Furthermore, this first-order upwinding method readily achieves a converged solution, while other RANS methods usually do not.

The LES model has other problems. The time-averaged LES flow is asymmetrical in the lower strand region after being simulated for 6 months, meaning that the simulation has not reached quasi-steady state due to 71 s simulation being too short. As the average residence time in the strand is 410 s, it would take ~ 3 years with current computational power to have a fully-developed LES flow, especially deep down the caster, which is needed for accurate particle simulations. Thus, the new model system with the chosen RANS model is used in the current study, as the best compromise of efficiency and accuracy.  

The above points have been added to the revised paper in section 5.

Point 8: Line 161; why is a 2-D heat conduction equation being solved for the mold wall? If the mold wall has a finite thickness and transverse gradients are important, a 3-D heat equation should have been solved.

Response 8: To answer this question, section 2.2 about CON1D model has been expanded. This model includes a one-dimensional transient finite-difference calculation of heat transfer and solidification within the steel shell coupled with a steady-state model of heat conduction within the mold wall, which captures the effects of the water slots across the width direction using a reduced order model (ROM). A paper describing this ROM model can be found in:

Lance C. Hibbeler, Melody M. Chin See, Junya Iwasaki, Kenneth E. Swartz, Ronald J. O'Malley, and Brian G. Thomas, “A Reduced-Order Model of Mold Heat Transfer in the Continuous Casting of Steel”, Applied Mathematical Modeling, 40, pp. 8530-8551, 2016. DOI.org/10.1016/j.apm.2016.04.002

The CON1D extends the ROM to a two-dimensional mold model near the meniscus, to account for the vertical heat flux there. CON1D can predict the shell thickness, temperature distributions in the mold and shell, heat transfer across the interfacial gap, and other related phenomena.

The current paper has been modified to reference the ROM paper. Please see lines 167-174.

Point 9: Please make sure to define the abbreviations first time they are used. Also, including them in figure captions makes it easier to understand the results.

Response 9: Thank you for the reminder. We certainly have intended to define every abbreviation at first use – if you spot one that we missed, please let us know.

As requested, both the definition and the abbreviation are added into every figure caption where needed.